# SELF-SUPERVISED DEBIASING USING LOW RANK REGULARIZATION

## ABSTRACT

Spurious correlations can cause strong biases in deep neural networks, impairing generalization ability. While most of existing debiasing methods require full supervisions on either spurious attributes or target labels, training a debiased model from a limited amount of both annotations is still an open issue. To overcome such limitations, we first examine an interesting phenomenon by the spectral analysis of latent representations: spuriously correlated, easy-to-learn attributes make neural networks inductively biased towards encoding lower effective rank representations. We also show that a rank regularization can amplify this bias in a way that encourages highly correlated features. Motivated by these observations, we propose a self-supervised debiasing framework that is potentially compatible with unlabeled samples. Specifically, we first pretrain a biased encoder in a self-supervised manner with the rank regularization, serving as a semantic bottleneck to enforce the encoder to learn the spuriously correlated attributes. This biased encoder is then used to discover and upweight bias-conflicting samples in a downstream task, serving as a boosting to effectively debias the main model. Remarkably, the proposed debiasing framework significantly improves the generalization performance of self-supervised learning baselines and, in some cases, even outperforms state-of-the-art supervised debiasing approaches.

## 1 INTRODUCTION

While modern deep learning solves several challenging tasks successfully, a series of recent works (Geirhos et al., 2018; Gururangan et al., 2018; Feldman et al., 2015) have reported that the high accuracy of deep networks on in-distribution samples does not always guarantee low test error on out-of-distribution (OOD) samples, especially in the context of spurious correlations. Arjovsky et al. (2019); Nagarajan et al. (2020); Tsipras et al. (2018) suggest that the deep networks can be potentially biased to the spuriously correlated attributes, or dataset bias, which are misleading statistical heuristics that are closely correlated but not causally related to the target label. In this regard, several recent works explain this phenomenon through the lens of simplicity bias (Rahaman et al., 2019; Neyshabur et al., 2014; Shah et al., 2020) of gradient descent-based deep networks optimization; deep networks prefer to rely on spurious features which are more "simpler" to learn, e.g., more linear.

The catastrophic pitfalls of dataset bias have facilitated the development of debiasing methods, which can be roughly categorized into approaches (1) leveraging annotations of spurious attributes, i.e., bias label (Kim et al., 2019; Sagawa et al., 2019; Wang et al., 2020; Tartaglione et al., 2021), (2) presuming specific type of bias, e.g., color and texture (Bahng et al., 2020; Wang et al., 2019; Ge et al., 2021) or (3) without using explicit kinds of supervisions on dataset bias (Liu et al., 2021; Nam et al., 2020; Lee et al., 2021; Levy et al., 2020; Zhang et al., 2022).

While substantial technical advances have been made in this regard, these approaches still fail to address the open problem: how to train a debiased classifier by fully exploiting unlabeled samples lacking *both* bias and target label. More specifically, while the large-scale unlabeled dataset can be potentially biased towards spuriously correlated sensitive attributes, e.g., ethnicity, gender, or age (Abid et al., 2021; Agarwal et al., 2021), most existing debiasing frameworks are not designed to deal with this unsupervised settings. Moreover, recent works on self-supervised learning have reported that self-supervised learning may still suffer from poor OOD generalization (Geirhos et al.,

2020; Chen et al., 2021; Robinson et al., 2021; Tsai et al., 2021) when such dataset bias still remains after applying data augmentations.

To address this question, we first made a series of observations about the dynamics of representations complexity by controlling the degree of spurious correlations in synthetic simulations. Interestingly, we found that spurious correlations suppress the effective rank (Roy & Vetterli, 2007) of latent representations, which severely deteriorates the semantic diversity of representations and leads to the degradation of feature discriminability. Another notable aspect of our findings is that the intentional increase of feature redundancy leads to amplifying "prejudice" in neural networks. To be specific, as we enforce the correlation among latent features to regularize the effective rank of representations (i.e., rank regularization), the accuracy on bias-conflicting samples quickly declines while the model still performs reasonably well on the bias-aligned [1] samples.

Inspired by these observations, we propose a self-supervised debiasing framework that can fully utilize potentially biased unlabeled samples. We pretrain (1) a biased encoder with rank regularization, which serves as a semantic bottleneck limiting the semantic diversity of feature components, and (2) the main encoder with standard self-supervised learning approaches. Specifically, the biased encoder gives us the leverage to uncover spurious correlations and identify bias-conflicting training samples in a downstream task.

**Contributions.** In summary, the contributions of this paper are as follows: First, we empirically demonstrate the inductive bias of neural networks in favor of low rank representations in the presence of spurious correlations. Based on these observations, we propose a novel rank-regularization debiasing framework that fully exploits unlabeled samples that do not contain annotation for bias and target label. Various experiments on real-world biased datasets demonstrate that retraining linear classifier in the last layer with upweighting of identified bias-conflicting samples significantly improves the OOD generalization in the linear evaluation protocol (Oord et al., 2018), even without making any modifications on the pretrained encoder. Our approach improves the accuracy on bias-conflicting evaluation set by $36.4\% \rightarrow 59.5\%$, $48.6\% \rightarrow 58.4\%$ on UTKFace (Zhang et al., 2017) and CelebA (Liu et al., 2015) with age and gender bias, respectively, compared to the best self-supervised baseline. Moreover, we found that the proposed framework outperforms state-of-the-art supervised debiasing methods in semi-supervised learning problem with CelebA.

## 2 LOW-RANK BIAS OF BIASED REPRESENTATIONS

### 2.1 PRELIMINARIES

**Preliminaries.** To evaluate the semantic diversity of given representation matrix, we introduce *effective rank* (Roy & Vetterli, 2007) which is a widely used metric to measure the effective dimensionality of matrix and analyze the spectral properties of features in neural networks (Arora et al., 2019; Razin & Cohen, 2020; Huh et al., 2021; Baratin et al., 2021):

**Definition 2.1** *Given the matrix $X \in \mathbb{R}^{m \times n}$ and its singular values $\{\sigma_i\}_{i=1}^{\min(m,n)}$, the effective rank $\rho$ of $X$ is defined as the shannon entropy of normalized singular values:*

$$\rho(X) = - \sum_{i=1}^{\min(m,n)} \bar{\sigma}_i \log \bar{\sigma}_i, \tag{1}$$

*where $\bar{\sigma}_i = \sigma_i / \sum_k \sigma_k$ is $i$-th normalized singular value. Without loss of generality, we omit the exponentiation of $\rho(X)$ as done in Roy & Vetterli (2007).*

Effective rank is also referred to as spectral entropy where its value is maximized when the singular values are all equal, and minimized when a top singular value dominates relative to all others. Recent works (Chen et al., 2019b;a) have revealed that the discriminability of representations resides on wide range of eigenvectors since the rich discriminative information for the classification

---

[1]The *bias-aligned* samples refer to data with a strong correlation between (potentially latent) spurious features and target labels. The *bias-conflicting* samples refer to the opposite cases where spurious correlations do not exist.

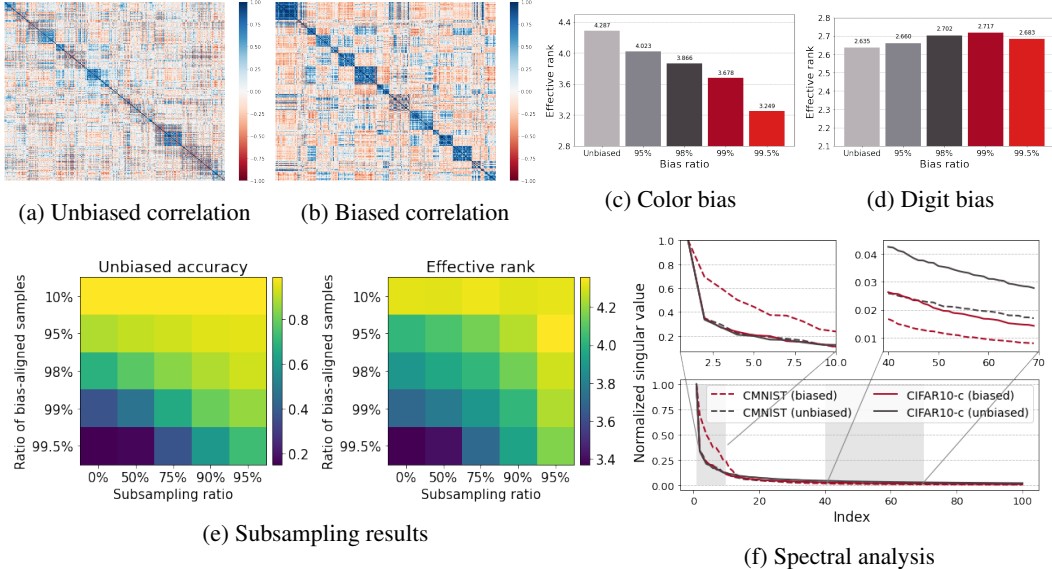

(a) Unbiased correlation     (b) Biased correlation     (c) Color bias     (d) Digit bias

(e) Subsampling results     (f) Spectral analysis

Figure 1: Empirical analysis on rank reduction phenomenon. (**a**, **b**): Hierarchically clustered auto-correlation matrix of unbiased and biased representations (Bias ratio=99%). (**c**, **d**): Effective rank with treating color or digit as dataset bias, respectively. 'Unbiased' represents the case training model with perfectly unbiased dataset, i.e., assign random color for each training sample. (**e**): Unbiased test accuracy (left) and effective rank (right) measured with subsampling bias-aligned samples. Subsampling ratio denotes the ratio of removed samples among the total bias-aligned samples. (**f**): SVD analysis with max-normalized singular values. Top 100 values are shown in the figure (Total: 256). All the analyses use the output $Z$ of the encoder (see notations in section 2.3).

task cannot be transmitted by only few eigenvectors with top singular values. Thus from a spectral analysis perspective, effective rank quantifies how diverse the semantic information encoded by each eigenfeature are, which is closely related to the feature discriminability across target label categories. In the rest of paper, we interchangeably use effective rank and rank by following prior works.

## 2.2 SPECTRAL ANALYSIS OF THE BIAS-RANK RELATIONSHIPS

**Degree of spurious correlations.** We now present experiments showing that the deep networks may tend to encode lower rank representations in the presence of stronger spurious correlations. To arbitrarily control the degree of spurious correlations, we introduce synthetic biased datasets, Color-MNIST (CMNIST) and Corrupted CIFAR-10 (CIFAR-10C) (Hendrycks & Dietterich, 2019), with color and corruption bias types, respectively. We define the degree of spurious correlations as the ratio of bias-aligned samples included in the training set, or bias ratio, where most of the samples are bias-aligned in the context of strong spurious correlations. Figure 1c shows that the rank of latent representations from a penultimate layer of the classifier decreases as the bias ratio increases in CMNIST. We provide similar rank reduction results of CIFAR-10C in the supplementary material.

We further compare the correlation matrix of biased and unbiased latent representations in the penultimate layer of biased and unbiased classifiers, respectively. In Figure 1a and 1b, we observe that the block structure in the correlation matrix is more evident in the biased representations after the hierarchical clustering, indicating that the features become highly correlated which may limit the maximum information capacity of networks.

We also measure the effective rank by varying the subsampling ratio (Japkowicz & Stephen, 2002) of bias-aligned samples. Subsampling controls the trade-off between the dataset size and the ratio of bias-conflicting samples to bias-aligned samples, i.e., conflict-to-align ratio, where subsampling of bias-aligned samples reduces the dataset size but increases the conflict-to-align ratio. Figure

1e shows that the effective rank is aligned well with the conflict-to-align ratio or generalization performance, whereas it is not along with the dataset size.

**Simplicity bias.** Here, we argue that the rank reduction is rooted in the simplicity bias (Shah et al., 2020; Hermann & Lampinen, 2020) of deep networks. Specifically, we reverse the task where the color is now treated as a target variable of networks, and the digit is spuriously correlated to the color, as done in (Nam et al., 2020). Digits are randomly assigned to color in an unbiased evaluation set. Figure 1d shows that the rank reduction is not reproduced in this switched context where the baseline levels of effective rank are inherently low. It intuitively implies that the rank reduction is evidence of reliance on easier-to-learn features, where the rank does not decrease progressively if the representation is already sufficiently simplified.

**Spectral analysis.** To investigate the rank reduction phenomenon in-depth, we compare the normalized singular values of biased and unbiased representations. Specifically, we conduct singular value decomposition (SVD) on the feature matrices of both biased and unbiased classifiers and plot the singular values normalized by the spectral norm of the corresponding matrix. Figure 1f shows that the top few normalized singular values of biased representations are similar to or even greater than that of unbiased representations. However, the remaining majority of singular values decay significantly faster in biased representations, greatly weakening the informative signals of eigenvectors with smaller singular values and deteriorating feature discriminability (Chen et al., 2019b;a).

## 2.3 RANK REGULARIZATION

Motivated from the rank reduction phenomenon, we ask an opposite-directional question: "Can we intentionally amplify the prejudice of deep networks by *maximizing* the redundancy between the components of latent representations?". If the feature components are extremely correlated, the corresponding representations may exhibit most of its spectral energy along the direction of one singular vector. For this case, effective rank may converge to 0. In other words, our goal is to design a *semantic bottleneck* of representations which restricts the semantic diversity of feature vectors. To implement the bottleneck in practice, we compute the auto-correlation matrix of the output of encoder. Throughout the paper, we denote $x \in \mathbb{R}^m$ and $y \in \mathcal{Y}$ as $m$-dimensional input sample and its corresponding predicting label, respectively. Then we denote $X = \{x_k\}_{k=1}^n$ as a batch of $n$ samples from a dataset which is fed to an encoder $f_\theta : \mathbb{R}^m \rightarrow \mathbb{R}^d$, parameterized by $\theta$. Then we construct a matrix $Z \in \mathbb{R}^{n \times d}$ where each $i$th row is the output representations of the encoder $f_\theta(x_i)^T$ for $x_i \in X$. Let $\bar{Z}$ denotes the mean-centered $Z$ along the batch dimension. The normalized auto-correlation matrix $C \in \mathbb{R}^{d \times d}$ of $\bar{Z}$ is defined as follow:

$$C_{i,j} = \frac{\sum_{b=1}^n \bar{Z}_{b,i} \bar{Z}_{b,j}}{\sqrt{\sum_{b=1}^n \bar{Z}_{b,i}^2} \sqrt{\sum_{b=1}^n \bar{Z}_{b,j}^2}} \quad \forall 1 \le i, j \le d, \tag{2}$$

where $b$ is an index of sample and $i, j$ are index of each vector dimension. Then we define our regularization term as negative of a sum of squared off-diagonal terms in $C$:

$$\ell_{reg}(X; \theta) = -\sum_i \sum_{j \neq i} C_{i,j}^2, \tag{3}$$

where we refer to it as a rank loss. Note that the target labels on $X$ is not used at all in formulation.

To investigate the impacts of rank regularization, we construct the classification model by combining the linear classifier $f_W : \mathbb{R}^d \rightarrow \mathbb{R}^c$ parameterized by $W \in \mathcal{W}$ on top of the encoder $f_\theta$, where $c = |\mathcal{Y}|$ is the number of classes. Then we trained models by cross entropy loss $\ell_{CE}$ combined with $\lambda_{reg} \ell_{reg}$, where $\lambda_{reg} > 0$ is a Lagrangian multiplier. We use CMNIST, CIFAR-10C, and Waterbirds dataset (Wah et al., 2011), and evaluate the trained models on an unbiased test set following Nam et al. (2020); Lee et al. (2021). After training models with varying the hyperparameter $\lambda_{reg}$, we compare bias-aligned and bias-conflict accuracy, which are the average accuracy on bias-aligned and bias-conflicting samples in the unbiased test set, respectively, for CMNIST and CIFAR-10C. Test accuracy on every individual data group is reported for Waterbirds.

Figure 2 shows that models suffer more from poor OOD generalization as trained with larger $\lambda_{reg}$. The average accuracy on bias-conflicting groups is significantly degraded, while the accuracy on bias-aligned groups is maintained to some extent. It implies that rank regularization may force

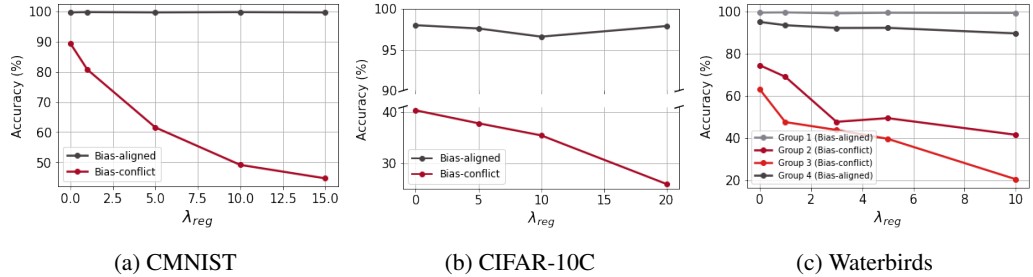

(a) CMNIST           (b) CIFAR-10C          (c) Waterbirds

Figure 2: (**a**, **b**): Bias-conflict and Bias-aligned accuracy on CMNIST and CIFAR-10C (Bias ratio=95%). (**c**): Group accuracy on Waterbirds. Detailed simulation settings are in the appendix.

|  | Precision (%) | Recall (%) |  | Precision (%) | Recall (%) |
|---|---|---|---|---|---|
| ERM | 85.59 | 19.76 | ERM | 52.03 | 0.06 |
| + Rank reg | **98.83** | **95.91** | + Rank reg | **71.39** | **51.43** |
| (a) CMNIST | | | (b) CIFAR-10C | | |

Table 1: Precision and recall of identified bias-conflicting samples in error set of ERM model trained with and without rank regularization. Bias ratio=95% for both dataset. $\lambda_{reg} = 35$ and $\lambda_{reg} = 20$ are used for CMNIST and CIFAR-10C, respectively. Capacity control techniques (e.g., strong $\ell_2$ regularization, early-stopping, Liu et al. (2021); Sagawa et al. (2019)) are not used to emphasize the contribution of rank regularization. Detailed simulation settings are in the appendix.

deep networks to focus on spurious attributes. Table 1 supports that the biased models with strong regularization can effectively probe out the bias-conflicting samples in the training set. Specifically, we train a biased classifier with rank regularization and distill an error set $E$ of misclassified training samples as bias-conflicting samples proxies. As reported in Table 1, we empirically observe that our biased classifier is relatively robust to the unintended memorization of bias-conflicting samples (Sagawa et al., 2020) in contrast to the standard models trained by Empirical Risk Minimization (ERM).

## 3 DEFUND: DEBIASING FRAMEWORK WITH UNLABELED DATA

Motivated by the observations in Section 2, we propose a self-supervised debiasing framework with unlabeled data, coined DeFund. The most important difference from prior works is that the proposed framework can intentionally learn biased representations without human supervision. Recent methods (Bahng et al., 2020; Nam et al., 2020; Liu et al., 2021; Lee et al., 2021; Zhang et al., 2022) train a biased model to uncover the spurious correlations and guide the main model to focus on samples that the biased model struggles to predict, which are seemingly the ones conflicting with the bias. While these methods require a bias label or a target label to train biased representations, we obtain such biased representations for free using self-supervised learning and rank regularization.

The proposed framework is composed of two stages: We first train the biased encoder, which can be potentially adopted to detect the bias-conflicting samples in a downstream task, along with the main encoder by self-supervised learning, both without any labels. After pretraining, we identify the bias-conflicting samples in the downstream task using linear evaluation protocol (Oord et al., 2018; Chen et al., 2020). This set of samples serves as a boosting to debias the main model.

We denote $f_\theta^{bias} : \mathcal{X} \to \mathbb{R}^d$ and $f_\phi^{main} : \mathcal{X} \to \mathbb{R}^d$ as biased encoder and main encoder parameterzied by $\theta \in \Theta$ and $\phi \in \Theta$, respectively, where $d$ is the dimensionality of latent representations. Then we can compute the rank loss in (3) with introduced encoders and given batch $\{x_k\}_{k=1}^N$ with size $N$. Let $f_{W_b}^{cls} : \mathbb{R}^d \to \mathbb{R}^C$ be a single-layer classifier parameterized by $W_b \in \mathcal{W}$ which is placed on top of biased encoder $f_\theta^{bias}$, where $C = |\mathcal{Y}|$ is the number of classes. We similarly define the linear classifier $f_{W_m}^{cls}$ for the main encoder. Then we refer to $f^{bias} : \mathcal{X} \to \mathbb{R}^C$ as biased

model, where $f^{bias}(x) = f^{cls}_{W_b}\left(f^{bias}_\theta(x)\right), \forall x \in \mathcal{X}$. We similarly define the main model $f^{main}$ as $f^{main}(x) = f^{cls}_{W_m}\left(f^{main}_\phi(x)\right), \forall x \in \mathcal{X}$. While the projection networks (Chen et al., 2020; Khosla et al., 2020) are employed as well, we omit the notations because they are not engaged in the linear evaluation after pretraining encoders.

**Stage 1. Train biased encoder.** To train the biased encoder $f^{bias}_\theta$, we revisit the proposed rank regularization term in (3) which can control the effective dimensionality of representations for instance discrimination task. We conjecture that the scope of captured features may be restricted to the easy-to-learn ones if the maximum information capacity of the encoder is strongly suppressed. Based on these intuitions, we apply rank regularization directly to the output of the base encoder, which encourages each feature component to be highly correlated. A simple simulation on the synthetic dataset conceptually clarifies the validity of our intuition, where we figured out that the representation becomes more biased as it is trained with stronger regularization, by measuring the bias metric, which quantifies how much the encoder focus on the short-cut attributes (Details provided in supplementary material). Moreover, while the overall performance may be upper-bounded due to the constraint on effective dimensionality (Jing et al., 2021), we observed that the bias-conflict accuracy is primarily sacrificed compared to the bias-aligned accuracy (Related experiments in Section 4).

**Stage 2. Debiasing downstream tasks.** After training the biased encoder, our next goal is to debias the main model, pretrained on the same dataset with standard self-supervised learning approaches, e.g., Chen et al. (2020); Chen & He (2021). To achieve this, we recall the recent work which explains the contrastive learning as a protocol inverting the data generating process; Zimmermann et al. (2021) demonstrates that the pretrained encoder with a contrastive loss from the InfoNCE family can recover the true latent factors of variation under some statistical assumptions. That being said, imagine that we have an ideal pretrained encoder of which each output component corresponds to the latent factor of data variation. Then one may expect that this encoder perfectly fits downstream classification tasks, where the only remaining job is to find out the optimal weights of these factors for prediction. However, if most samples in the downstream task are bias-aligned, these samples may misguide the model to upweight the spuriously correlated latent factors. In other words, the model may reach a biased solution even though it encodes well-generalized representations.

The above contradiction elucidates the importance of bias-conflicting samples, which serve as counterexamples of spuriously correlated feature components, thereby preventing the alleged involvement of such components in prediction. Based on these intuitions, we introduce a novel debiasing protocol that probes and upweights bias-conflicting samples to find and fully exploit feature components independent of spurious correlations. We evaluate our framework on two scenarios: linear evaluation and semi-supervised learning. First, following the conventional protocol of self-supervised learning, we conduct linear evaluation (Zhang et al., 2016; Oord et al., 2018), which trains a linear classifier on top of unsupervised pretrained representations by using target labels of every training sample. After training a linear classifier $f^{cls}_{W_b}$ with pretrained biased encoder $f^{bias}_\theta$ given the whole training set $D = \{(x_k, y_k)\}_{k=1}^N$ with size $N$, an error set $E$ of misclassified samples and corresponding labels is regarded as bias-conflicting pairs. Then we train a linear classifier $f^{cls}_{W_m}$ on freezed representations of main encoder $f^{main}_\phi$ by upweighting the identified samples in $E$ with $\lambda_{up} > 0$. The loss function for *debiased* linear evaluation is defined as follow:

$$\ell_{debias}(D; W_m) = \lambda_{up} \sum_{(x,y) \in E} \ell(x, y; W_m) + \sum_{(x,y) \in D \setminus E} \ell(x, y; W_m), \qquad (4)$$

where we use cross entropy loss for $\ell : \mathcal{X} \times \mathcal{Y} \times \mathcal{W} \to \mathbb{R}^+$. Note that the target labels are only used in training linear classifiers after pretraining.

While linear evaluation is mainly opted for evaluating self-supervised learning methods, we also compare our method directly to other supervised debiasing methods in the context of semi-supervised learning. Here we assume that the training dataset includes only a small amount of labeled samples combined with a large amount of unlabeled samples. As in linear evaluation, we train a linear classifier on top of the biased encoder by using labeled samples. After obtaining an error set $E$ of misclassified samples, we finetuned the whole main model by upweighting the identified samples in $E$ with $\lambda_{up}$. Note that supervised baselines are restricted to using only a small fraction of labeled samples, while the proposed approach benefits from the abundant unlabeled samples during training of the biased encoder. The pseudo-code of DeFund is provided in the supplementary material.

Table 2: (Linear evaluation) Bias-conflict and unbiased test accuracy (%) evaluated on UTKFace and CelebA. Models requiring information on target class or dataset bias in (pre)training stage are denoted with ✓in column Y and B, respectively. The results are averaged on 4 random seeds.

| Model | Y | B | UTKFace (age) | | UTKFace (gender) | | CelebA (makeup) | |
|---|---|---|---|---|---|---|---|---|
| | | | Conflict | Unbiased | Conflict | Unbiased | Conflict | Unbiased |
| LNL | ✓ | ✓ | $45.8_{\pm0.6}$ | $72.6_{\pm0.3}$ | $73.1_{\pm1.6}$ | $84.9_{\pm0.8}$ | $55.9_{\pm2.1}$ | $76.0_{\pm0.6}$ |
| EnD | ✓ | ✓ | $45.3_{\pm0.9}$ | $72.2_{\pm0.2}$ | $75.5_{\pm1.1}$ | $85.5_{\pm0.4}$ | $57.3_{\pm2.4}$ | $76.4_{\pm1.4}$ |
| JTT | ✓ | ✗ | $63.8_{\pm0.9}$ | $69.4_{\pm1.3}$ | $71.2_{\pm0.3}$ | $77.6_{\pm0.4}$ | $62.4_{\pm1.2}$ | $74.7_{\pm0.8}$ |
| CVaR DRO | ✓ | ✗ | $45.7_{\pm2.0}$ | $71.4_{\pm0.3}$ | $68.6_{\pm1.0}$ | $81.0_{\pm0.8}$ | $58.0_{\pm1.7}$ | $76.5_{\pm0.6}$ |
| ERM | ✓ | ✗ | $45.4_{\pm2.1}$ | $71.0_{\pm1.2}$ | $65.7_{\pm1.4}$ | $79.5_{\pm0.6}$ | $54.2_{\pm0.2}$ | $74.1_{\pm1.4}$ |
| SimSiam | ✗ | ✗ | $28.2_{\pm0.9}$ | $62.6_{\pm0.7}$ | $48.5_{\pm1.0}$ | $69.8_{\pm0.7}$ | $39.9_{\pm0.6}$ | $66.7_{\pm0.6}$ |
| VICReg | ✗ | ✗ | $32.3_{\pm0.6}$ | $64.6_{\pm0.3}$ | $51.0_{\pm1.4}$ | $71.3_{\pm0.7}$ | $48.6_{\pm0.6}$ | $71.9_{\pm0.2}$ |
| SimCLR | ✗ | ✗ | $36.4_{\pm1.5}$ | $66.3_{\pm0.6}$ | $56.3_{\pm0.2}$ | $74.2_{\pm0.2}$ | $46.9_{\pm1.0}$ | $69.8_{\pm0.4}$ |
| **DeFund** | ✗ | ✗ | $\mathbf{59.5}_{\pm0.8}$ | $\mathbf{70.6}_{\pm0.8}$ | $\mathbf{63.7}_{\pm2.0}$ | $\mathbf{74.9}_{\pm0.9}$ | $\mathbf{58.4}_{\pm0.6}$ | $\mathbf{73.1}_{\pm1.0}$ |

## 4 RESULTS

### 4.1 METHODS

**Dataset.** To investigate the effectiveness of the proposed debiasing framework, we evaluate several supervised and self-supervised baselines on UTKFace (Zhang et al., 2017) and CelebA (Liu et al., 2015) in which prior work has observed poor generalization performance due to spurious correlations. Each dataset includes several sensitive attributes, e.g., gender, age, ethnicity, etc. We consider three prediction tasks: For UTKFace, we conduct binary classifications using (Gender, Age) and (Race, Gender) as (target, spurious) attribute pair, which we refer to UTKFace (age) and UTKFace (gender), respectively. For CelebA, we consider (HeavyMakeup, Male) and (Blonde Hair, Male) as (target, spurious) attribute pairs, which are referred to CelebA (makeup) and CelebA (blonde), respectively. The results of CelebA (blonde) are reported in appendix. Following Nam et al. (2020); Hong & Yang (2021), we report bias-conflict accuracy together with unbiased accuracy, which is evaluated on the explicitly constructed validation set. We exclude the dataset in Figure 2 based on the observations that the SimCLR models are already invariant w.r.t spurious attributes.

**Baselines.** We mainly target baselines consisting of recent advanced self-supervised learning methods, SimCLR (Chen et al., 2020), VICReg (Bardes et al., 2021), and SimSiam (Chen & He, 2021), which can be categorized into contrastive (SimCLR) and non-contrastive (VICReg, SimSiam) methods. We further report the performance of vanilla networks trained by ERM, and other supervised debiasing methods such as LNL (Kim et al., 2019), EnD (Tartaglione et al., 2021), and upweighting-based algorithms, JTT (Liu et al., 2021) and CVaR DRO (Levy et al., 2020), which can be categorized into methods that leverage annotations on dataset bias (LNL, EnD) or not (JTT, CVaR DRO).

**Optimization setting.** Both bias and main encoder is pretrained with SimCLR (Chen et al., 2020) for 100 epochs on UTKFace, and 20 epochs on CelebA, respectively, using ResNet-18, Adam optimizer and cosine annealing learning rate scheduling (Loshchilov & Hutter, 2016). We use a MLP with one hidden layer for projection networks as in SimCLR. All the other baseline results are reproduced by tuning the hyperparameters and optimization settings using the same backbone architecture. We report the results of the model with the highest bias-conflicting test accuracy over those with improved unbiased test accuracy compared to the corresponding baseline algorithms, i.e., SimCLR for ours. The same criteria are applied to supervised baselines, while JTT often sacrifices unbiased accuracy for highly improved bias-conflict accuracy. More details about the dataset and simulation settings are provided in the supplementary material.

### 4.2 EVALUATION RESULTS

**Linear evaluation.** The bias-conflict and unbiased test accuracy are summarized in Table 2. We found that DeFund outperforms every self-supervised baseline by a large margin, including Sim-CLR, SimSiam and VICReg, with respect to both bias-conflict and unbiased accuracy. Moreover, in

Table 3: (Semi-supervised learning) Bias-conflict and unbiased accuracy (%) evaluated on CelebA (makeup). Label fraction is set to 10%. Each first and second ✓marker represents whether the model requires information on target class or dataset bias in pretraining stage, respectively.

| Accuracy | LNL ✓✓ | EnD ✓✓ | JTT ✓✗ | CVaR DRO ✓✗ | ERM ✓✗ | SimCLR ✗✗ | DeFund ✗✗ |
|---|---|---|---|---|---|---|---|
| Conflict | $55.7_{\pm1.4}$ | $55.3_{\pm1.5}$ | $51.5_{\pm1.9}$ | $55.6_{\pm1.5}$ | $51.5_{\pm1.1}$ | $50.5_{\pm4.7}$ | $\mathbf{60.5}_{\pm0.4}$ |
| Unbiased | $75.6_{\pm0.5}$ | $\mathbf{76.2}_{\pm0.8}$ | $71.4_{\pm1.3}$ | $75.7_{\pm1.0}$ | $73.1_{\pm0.3}$ | $71.6_{\pm1.9}$ | $75.6_{\pm0.2}$ |

some cases, DeFund even outperforms ERM models or supervised debiasing approaches regarding bias-conflict accuracy. Note that there is an inherent gap between ERM models and self-supervised baselines, roughly 8.7% on average. Moreover, we found that non-contrastive learning methods generally perform worse than the contrastive learning method. This warns us against training the main model using a non-contrastive learning approach, while it may be a viable option for the biased model. We provide results of the proposed framework implemented with non-contrastive learning methods in the supplementary material.

**Semi-supervised learning.** To compare the performance of supervised and self-supervised methods in a more fair scenario, we sample 10% of the labeled CelebA training dataset at random for each run. The remaining 90% samples are treated as unlabeled ones and engaged only in pretraining encoders for self-supervised baselines. Labeled samples are provided equally to both supervised and self-supervised methods.

Remarkably, Table 3 shows that the proposed framework outperforms all the other state-of-the-art supervised debiasing methods. Notably, only about 16 samples remain within (`Gender=1`, `HeavyMakeup=1`) group after subsampling. Thus it is almost impossible to prevent deep networks from memorizing those samples even with strong regularization if we train the networks from scratch, which explains the failure of existing upweighting protocols such as JTT. In contrast, the proposed framework can fully take advantage of unlabeled samples where contrastive learning help prevent memorization of the minority counterexamples (Xue et al., 2022). It highlights the importance of pretraining using unlabeled samples that most prior debiasing works do not consider. Moreover, such implicit bias of deep networks towards memorizing samples may seriously deteriorate the performance of existing bias-conflicting sample mining algorithms (Kim et al., 2021; Zhao et al., 2021; Nam et al., 2020) when the number of labeled samples is strictly limited. However, such failure is unlikely to be reproduced in the proposed framework since we only train a simple linear classifier on top of a freezed biased encoder to identify such bias-conflicting samples.

Table 4: Ablation study on introduced modules. Accuracy is reported in (%).

| Method | UTKFace (age) | | UTKFace (gender) | | CelebA (makeup) | |
|---|---|---|---|---|---|---|
| | Conflict | Unbiased | Conflict | Unbiased | Conflict | Unbiased |
| SimCLR | 36.4 | 66.3 | 56.3 | 74.2 | 46.9 | 69.8 |
| + Rank reg | 26.6 | 61.3 | 50.9 | 70.3 | 43.9 | 68.3 |
| + Upweight | 53.0 | 64.6 | 58.3 | 74.5 | 50.1 | 70.4 |
| **DeFund** | **59.5** | **70.6** | **63.7** | **74.9** | **58.4** | **73.1** |

Table 5: Precision and recall (%) of bias-conflicting samples identified by SimCLR and our biased model. Both case used linear evaluation.

| Method | UTKFace (age) | | UTKFace (gender) | | CelebA | |
|---|---|---|---|---|---|---|
| | Precision | Recall | Precision | Recall | Precision | Recall |
| SimCLR | 68.31 | 44.63 | **33.36** | 39.59 | 52.25 | 28.23 |
| **DeFund** | **68.67** | **75.94** | 29.98 | **50.93** | **55.29** | **32.46** |

**Ablation study.** To quantify the extent of performance improvement achieved by each introduced module, we compared the linear evaluation results of (a) vanilla SimCLR, (b) SimCLR with rank regularization, (c) SimCLR with upweighting error set $E$ of the main model, and (d) Full model DeFund. Note that (c) does not use a biased model at all. Table 4 shows that every module plays an important role in OOD generalization. Considering that the main model is already biased to some extent, we found that bias-conflict accuracy can be improved even without a biased model, where the error set $E$ of the biased model further boosts the generalization performance. We also quantify how well the biased model captures bias-conflicting samples by measuring the precision and recall of identified bias-conflicting samples in $E$. As reported in Table 5, the biased model detects more diverse bias-conflicting samples compared to the baseline for free or with affordable precision costs. While the improvement of recall in CelebA may seem relatively marginal, a large quantity of bias-conflicting samples is additionally identified in practice considering that CelebA includes much more samples than UTKFace.

## 5 DISCUSSIONS AND CONCLUSION

**Contributions.** In this paper, we (**a**) first unveil the catastrophic adverse impacts of spurious correlations on the effective dimensionality of representations. Based on these findings, we (**b**) design a rank regularization that amplifies the feature redundancy by reducing the spectral entropy of latent representations. Then we (**c**) propose a debiasing framework empowered by the biased model pretrained with abundant unlabeled samples.

**Comparisons to related works.** Our observations are in line with the simplicity bias of gradient descent-based optimizations, where many recent studies (Rahaman et al., 2019; Shah et al., 2020) have revealed that networks tend to exploit the simplest feature at the expense of a small margin and often ignore the complex features. Similar observations have been made confined to self-supervised learning named feature suppression, where the encoder may heavily rely on the attributes that make the instance discrimination tasks easier. While these existing works often focus on the innate preference of models on input cues (Hermann & Lampinen, 2020; Scimeca et al., 2021), we provide a novel perspective on the practical impacts of spurious correlations on deep latent representations: reduction of effective rank.

Robinson et al. (2021) proposes an opposite-directional approach compared to our framework to improve generalizations of self-supervised learning. It aims to overcome the feature suppression and learn a wide variety of features by Implicit Feature Modification (IFM), which adversarially perturbs feature components of the current representations used to discriminate instances, thereby encouraging the encoder to use other informative features. We observed that IFM improves the bias-conflict accuracy by about 1% on UTKFace (age) in Table 6, which is roughly consistent with the performance gains on the standard benchmarks, e.g., STL10, reported in the original paper. However, its performance gain is relatively marginal compared to the proposed framework.

Table 6: Results of Implicit Feature Modification (Robinson et al., 2021) with SimCLR on UTKFace (age). we denote $\epsilon$ as the adversarial budget of feature modification as in the original paper.

| Accuracy | SimCLR | $\epsilon = 0.05$ | $\epsilon = 0.1$ | $\epsilon = 0.5$ |
|---|---|---|---|---|
| Bias-conflict (%) | 36.4 | **37.5** | 36.4 | 33.7 |
| Unbiased (%) | 66.3 | **66.5** | 66.2 | 64.6 |

**Future directions.** While this work has focused on intentionally encoding biased representations, we argue that more advances should be concurrently made in learning both biased and debiased representations, as partially discussed above (Robinson et al., 2021). The interplay between those bidirectional modules may further improve generalizations. We also note that the proposed rank regularization is one possible implementation of the semantic bottleneck. While we explicitly control the feature correlations, we believe such design can be employed more implicitly. We provide experiments examining the potential of existing hyperparameters, e.g., the temperature in InfoNCE loss, etc., as a bias controller in the supplementary material. Lastly, we believe that a semi-supervised learning scenario should be part of a standard evaluation pipeline where many supervised baselines may fail due to the inductive bias of networks towards memorizing a few counterexamples.

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

# Appendix

The supplementary material is organized as follows. We begin with providing the algorithm of DeFund, followed by more discussions on the related works. Then we provide additional results and analyses in section C. Optimization setting, hyperparameter configuration and other experimental details are provided in section D.

## A  PSEUDOCODE

---

**Algorithm 1** Debiasing Framework with unlabeled data

---

1: **Input:** $D_l = \{(x_k, y_k)\}_{k=1}^{N_1}$, $D_u = \{x_k\}_{k=1}^{N_2}$ for semi-supervised learning ($N_2 \gg N_1$), or $\varnothing$ for linear evaluation, $D = D_l \cup D_u$, batch size $n$, structure of $f^{bias}$ and $f^{main}$.
2:
3: **Stage 1.** *Pretraining encoders*
4: **for** subsampled minibatch $X = \{x_k\}_{k=1}^n$ from D **do**
5:     Update $\theta$ of $f_\theta^{bias}$ with SimCLR NT-Xent loss and $\lambda_{reg}\ell_{reg}(X; \theta)$.
6:     Update $\phi$ of $f_\phi^{main}$ with SimCLR NT-Xent loss.
7: **end for**
8: Obtain pretrained parameters $\hat{\theta}$ and $\hat{\phi}$.
9:
10: **Stage 2.** *Downstream task*
11: Freeze $f_{\hat{\theta}}^{bias}$ and train $f_{W_b}^{cls}$ with $D_l$. Identify the error set $E \subset D_l$ with trained $f^{bias}$.
12: **if** Linear evaluation **then**
13:     Freeze $f_{\hat{\phi}}^{main}$ and train $f_{W_m}^{cls}$ with $\ell_{debias}(D_l; W_m)$
14: **else if** Semi-supervised learning **then**
15:     Finetune $f^{main}$ with $\ell_{debias}(D_l; W_m, \phi)$ where $\phi$ is initialized with $\hat{\phi}$.
16: **end if**

---

## B  SUPPLEMENTARY INFORMATION - RELATED WORKS

Here, we have detailed discussions about related works.

**Discovering bias without supervision.** In practice, several limitations exist against gleaning more labeled samples: labeling budget, expert-level knowledge required for labeling, data privacy, etc. In this regard, most training samples lack annotations on the spuriously correlated attributes.

To mitigate these problems, several works aim to discover biases without bias annotations. Liu et al. (2021) reveals that the standard ERM model may serve as a bias-capturing model if one trains it with strong capacity control. Yaghoobzadeh et al. (2019) shows that forgettables, or examples that have been forgotten at least once, contain more minority examples, and proposes a novel robust learning framework by fully exploiting the identified forgettable examples. Li & Xu (2021) obtains a biased attribute hyperplane of the generative models, which can help identify semantic biases by generating bias-traversal images. Li et al. (2022) introduces the discoverer model, which uncovers multiple unknown biases such that the difference of averaged predicted probabilities on the target attribute in two groups is maximized. Lang et al. (2021) proposes a novel framework, StylEx, which trains a styleGAN to specifically visualize multiple attributes underlying the classifier decisions.

While substantial advances have been made in discovering the unknown biases of neural networks without bias labels, these works still inevitably require target labels. In contrast, we consider a very challenging scenario that has received little attention so far: self-supervised debiasing. In this regard, our work addresses the following open problems/questions:

- Can we learn biased/debiased representations by using unlabeled samples?
- What is the fundamental difference between biased and debiased representations?

- Is supervised debiasing robust despite decreasing the number of labeled samples?
- How can bias-conflicting samples be discovered by leveraging information from unlabeled samples?
- Many recent works have reported the limitations of self-supervised learning (SSL) in OOD generalization. How can we overcome such limitations?

**Mitigating bias with reweighting.** Recently, Kirichenko et al. (2022) have reported an intriguing observation: Simple last layer retraining, so-called Deep Feature Reweighting (DFR), can match or outperform state-of-the-art approaches on spurious correlation benchmarks. Kirichenko et al. (2022) shows that biased classifiers still often learn core features associated with the desired attributes of the data. Based on these observations, they probe invariant features for the reweighting by leveraging explicit group-balanced dataset $\hat{D}$.

We compare the proposed framework with DFR as follows. First, while DFR and the proposed framework can mitigate the bias in representations by retraining the last linear layer, our method is not restricted to such last-layer retraining. Instead, the semi-supervised learning scenario is a more practical application of the proposed method. Specifically, we can fine-tune representations by fully exploiting both unlabeled and labeled samples, which improves the performance compared to the last layer retraining in Table 3. In contrast, DFR trains a linear classifier while freezing the pretrained representations as-is. More importantly, DFR requires pretrained networks or fully labeled datasets where we consider a more challenging scenario without such assumptions. Moreover, DFR does not use mining bias-conflicting samples in the training set. Specifically, DFR trains a new classification head from scratch on the available group-balanced data $\hat{D}$. In Kirichenko et al. (2022), the reweighting dataset $\hat{D}$ often consists of a random group-balanced subset of the training or validation data. In other words, DFR is not designed to identify the bias-conflicting samples but exploits the existing group annotations. Considering practical situation with several limitations against collecting more labeled samples, it remains unclear how to obtain the group-balanced dataset $\hat{D}$ with sufficient number of samples in the absence of prior information on the dataset bias. In contrast, the proposed framework can leverage the explicit set $\hat{D}$ if accessible, *as well as* identifying the unknown bias-conflicting samples in the training set.

## C  ADDITIONAL RESULTS

Our additional results can be roughly categorized into: (1) more observations related to the rank reduction, (2) rank regularization in self-supervised learning, and (3) an examination of the potential of existing hyperparameters as a bias controller. Our observations include the visualization of reconstructed images with biased representations, rank reduction trends in CIFAR-10C and Vision Transformer (ViT, Dosovitskiy et al. (2020)), minority mining performance in supervised settings, and rank regularization with a moderate level of bias. Then we present a simple synthetic simulation on the behavior of rank-regularized encoder, followed by additional results on non-contrastive methods and CelebA (blonde). Then the potential of using shallow networks as the bias-capturing model will be discussed. Lastly, we provide additional analysis on relations between existing hyperparameters of self-supervised learning and effective rank.

### C.1  MORE OBSERVATIONS

**Reconstruction of biased representations.** To understand the relationship between rank regularization and spurious correlations more deeply, we visualize the pretrained representations with varying degrees of bias. We first trained deep networks on: (a) unbiased CMNIST (random background color), (b) biased CMNIST (bias ratio=95%) without rank regularization and (c) with rank regularization ($\lambda_{reg} = 50$). Then, we train the auxiliary decoder, which reconstructs the bias-conflicting images from freezed latent representations of each pretrained network.

Figure 3 shows that the reconstructed results are evidently different for each case. First, the decoder successfully reconstructs the foreground digit from the (a) unbiased representations, while the background color is completely changed in some cases. It implies that unbiased representations may

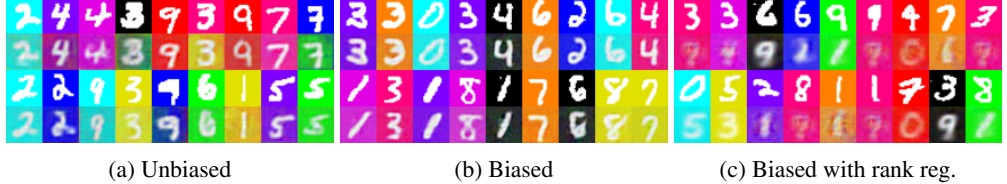

| (a) Unbiased | (b) Biased | (c) Biased with rank reg. |

Figure 3: Randomly selected reconstructed images from representations with varying degrees of bias. First and third row correspond to the input bias-conflicting images. Second and fourth row correspond to the reconstructed images. Reconstructed from (**a**) unbiased representations, (**b**) biased representations, and (**c**) biased representations with rank regularization (bias ratio=95% in **b, c**).

lack information on spuriously correlated attributes, i.e., background color. However, both digit and color are well reconstructed in (b) biased case, implying that the biased model encodes both spurious and invariant features. Intriguingly, the decoder fails to reconstruct bias-conflicting images from the (c) biased representations pretrained with rank regularization. Specifically, the foreground digit is blurred, and its class is often changed following the color-digit assignment in Figure 4.

Based on these observations, we summarize some key insights: First, the rank-regularized representation may lose its information on harder-to-learn invariant features. While the reconstructed images in (a) or (b) preserve the detailed class, shape, and style of the foreground digit, such properties are deteriorated in (c), implying the loss of feature discriminability and informative signals. Second, with limited semantic diversity, the rank-regularized model fails to identify the true underlying independent generative factors for multidimensional data; it may rather encode feature components entangled with both spurious and invariant attributes. In other words, the proposed low-rank regularization prevents features from encoding discriminative information independently. It is indirectly reflected in (c) that the digit class of the reconstructed image is erroneously determined by the spurious attribute, i.e., background color. The experiments showed that our rank regularization encourages the network to focus more on spurious correlations in a way that minimizes semantic diversity.

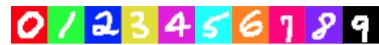

Figure 4: Examples of bias-aligned CMNIST images.

Table 7: Unbiased test accuracy (%) on CMNIST and CIFAR-10C measured with varying bias ratio $r$. The model trained with unbiased dataset ($r = 10\%$) serves as a baseline.

| Dataset | Unbiased | $r = 95\%$ | $r = 98\%$ | $r = 99\%$ | $r = 99.5\%$ |
|---------|----------|-----------|-----------|-----------|-------------|
| CMNIST | 99.87 | 88.27 | 68.13 | 36.21 | 13.61 |
| CIFAR-10C | 78.71 | 46.15 | 34.18 | 26.76 | 20.94 |

**Rank reduction.** Figure 5a shows that the rank of latent representations from a penultimate layer of classifier decreases as the bias ratio increases in CIFAR-10C. In Table 7, we supplement the unbiased test accuracy of CMNIST and CIFAR-10C from the experiments presented in Figure 1c and 5a, respectively. Moreover, similar rank reduction trends are observed in Vision Transformer (ViT, Dosovitskiy et al. (2020)). We train ViT on CMNIST and CIFAR-10C for 2000 and 10000 iterations, respectively, with Adam optimizer of learning rate 0.001, patch size 4, dimension of output tensor 128, number of transformer blocks 6, number of heads in multi-head Attention layer 4, dropout rate 0.2 and dimension of the MLP (FeedForward) layer 1024. Figure 5b, 5c show that the effective rank of the output of the Transformer encoder $\mathbf{z}_L^0$ (notation follows the original paper) decreases as bias ratio increases.

**Minority mining in supervised setting.** Following the official implementation of JTT (Liu et al., 2021), we compare the quality of bias-conflicting samples identified by JTT and the rank-regularized model in Waterbirds dataset. Following the official command of JTT, $\lambda_{\ell_2} = 1$ is used for JTT, while $\lambda_{\ell_2} = 1e-4$ is used for ours to focus on the contribution of rank regularization. The number of training epochs is selected by tuning $T \in \{20, 40, 60\}$ for both JTT ($T = 60$) and ours ($T = 40$). All the other experimental settings are fixed, including backbone architecture (ResNet-50) and optimizer

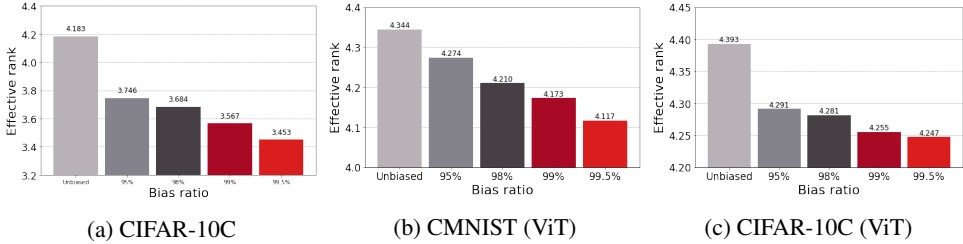

(a) CIFAR-10C      (b) CMNIST (ViT)      (c) CIFAR-10C (ViT)

Figure 5: Effective rank measured with (**a**) CIFAR-10C (ResNet-18), (**b**) CMNIST (ViT) and (**c**) CIFAR-10C (ViT).

(Adam), etc. Table 8 shows that the proposed rank regularization improves the precision and recall of identified bias-conflicting samples by using sufficiently large $\lambda_{reg}$. Even though these results imply that rank regularization may be a viable option for the minority mining algorithms in a supervised learning setting, we defer extensive discussions about this possibility to focus on the unsupervised learning settings.

Table 8: Precision and recall of bias-conflicting samples identified by ERM, JTT and the proposed biased model in Waterbirds dataset.

| Metrics | ERM | JTT | Ours ($\lambda_{reg} = 5$) | Ours ($\lambda_{reg} = 10$) |
|---|---|---|---|---|
| Precision (%) | 37.84 | 48.95 | 48.91 | **54.77** |
| Recall (%) | 11.67 | 48.75 | 46.67 | **55.01** |

**Rank regularization with moderate level of bias.** To study the compatibility of rank regularization with weak spurious correlations, we apply the rank regularization to the moderately biased CMNIST, i.e., bias ratio=60%. Table 9 shows that the rank regularization works well in this natural setting. This implies that the rank regularization can be leveraged to reveal the moderate level of bias embedded in the representations, which is supported by the empirical results of other general datasets, e.g., Waterbirds, UTKFace or CelebA.

Table 9: Ablation study of rank regularization on weakly biased CMNIST (Bias ratio=60%). Our rank-regularized model is trained with $\lambda_{reg} = 50$. For a fair comparison, all the other experimental settings are fixed. Bias-aligned accuracy, bias-conflict accuracy, precision and recall of identified bias-conflicting samples are reported.

| Methods | Align (%) | Conflict (%) | Precision (%) | Recall (%) |
|---|---|---|---|---|
| ERM | 99.49 | 97.81 | 79.55 | 0.87 |
| Ours | 96.25 | 38.15 | 91.56 | 60.97 |

**Behavior of rank-regularized encoder.** Here, we present a simple simulation which conceptually clarifies the impacts of rank regularization in self-supervised learning. Inspired from Chen et al. (2020); Robinson et al. (2021), we create a DigitsOnSTL10 dataset as in Figure 6a where MNIST images are randomly selected and placed on top of the STL10 images. After self-supervised representation learning, we train two independent linear classifiers on top of the frozen representations, where we provide label of foreground MNIST digit for one classifier, and label of background STL10 object class for the other. After training linear classifiers, we measure the ratio of MNIST classifier test accuracy to STL10 classifier test accuracy, which we treat as a proxy of ratio of spuriously correlated features to invariant features, i.e., degree of bias in representations. Intuitively, the proposed bias metric increases as the encoder focus more on the short-cut attribute, i.e., MNIST digit.

We measure the bias metric on the representations of ResNet-18 encoders trained by SimCLR (Chen et al., 2020) together with rank regularization loss $\lambda_{reg}\ell_{reg}$, where $\lambda_{reg} > 0$ is a balancing hyperparameter. As denoted in the main paper, we apply regularization not on the output of projection

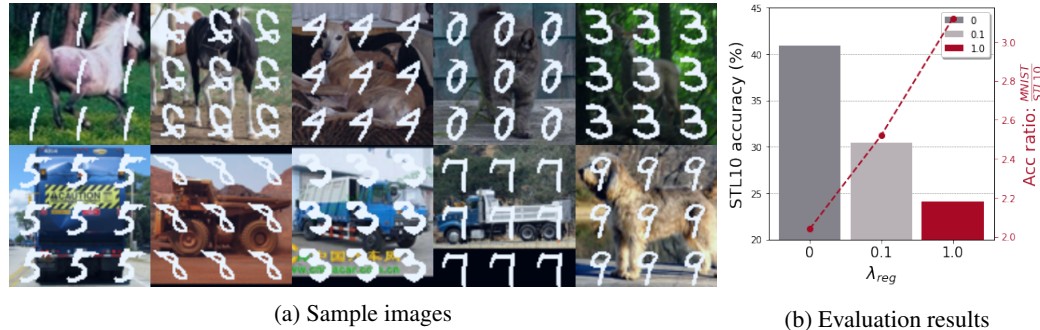

(a) Sample images              (b) Evaluation results

Figure 6: (**a**) Sample images from DigitsOnSTL10 dataset. (**b**) Test accuracy of STL10 classifier and bias metric.

networks but directly on the output of base encoder, which makes it fully agnostic to networks architecture. Figure 6b shows that the rank regularization exacerbates the "feature suppression" phenomenon revealed by Chen et al. (2021). The representation becomes more biased as it is trained with stronger regularization. While the overall performance of self-supervised learning may be upper-bounded due to the constraint on effective dimensionality (Jing et al., 2021), we observe in Figure 6b that the bias-conflict accuracy is primarily sacrificed compared to the bias-aligned accuracy. Coupled with results in section 4, this result implies that rank regularization can amplify bias in self-supervised encoder.

## C.2 ADDITIONAL RESULTS ON CELEBA

We report the results of CelebA (blonde) in here due to the limited space. Detailed information on the dataset and simulation settings is provided in the section D. Following Sagawa et al. (2019); Liu et al. (2021), we report worst-group and average accuracy because CelebA (blonde) includes abundant samples in (`Blonde Hair`=0, `Male`=0) bias-conflicting group. The number of training samples in each group is provided in Table 17.

Table 10 shows that DeFund outperforms not only every self-supervised baseline, but also ERM, CVaR DRO, and LfF Nam et al. (2020) in linear evaluation. Table 11 shows that DeFund outperforms all the other baseline methods in semi-supervised learning, which is consistent with Table 3 of the main paper.

Moreover, recent works unveil that CelebA (blonde) exhibits a large class imbalance which in turn correlates with a large group imbalance. Hong & Yang (2021); Idrissi et al. (2022) found that both target classes are biased toward a non-Male bias class in CelebA (blonde) which obfuscates whether the dataset is indeed biased. In this regard, Idrissi et al. (2022) observed that the simple class balancing serves as a powerful baseline due to the class imbalance. This directly motivates us to alleviate the class imbalance and focus on the dataset bias itself. Following Hong & Yang (2021), we randomly subsample images from (`Blonde Hair`=0, `Male`=0) group so that two target classes are biased toward different bias classes. The number of training samples before and after subsampling is provided in Table 17d and 12b, respectively. Table 12a shows that DeFund outperforms JTT with respect to both worst-group and average accuracy. These additional results imply that the proposed framework ensures reliable performance in the presence of strong spurious correlations.

## C.3 NON-CONTRASTIVE METHODS

We provide the results of proposed framework implemented based on non-contrastive methods. Specifically, we leverage SimSiam (Chen & He, 2021) and VICReg (Bardes et al., 2021) as baselines. Table 13 shows that the generalization performance of both baselines can be improved with the proposed debiasing framework.

Table 10: (Linear evaluation) Worst-group and average accuracy (%) evaluated on CelebA (blonde). Results of ERM, CVaR DRO, LfF (Nam et al., 2020) and JTT are come from Table 1 of the original JTT paper (Liu et al., 2021). Each first and second ✓marker represents whether the model requires information on target class or dataset bias in pretraining stage, respectively.

| Accuracy | ERM ✓✗ | CVaR DRO ✓✗ | LfF ✓✗ | JTT ✓✗ | VICReg ✗✗ | SimSiam ✗✗ | SimCLR ✗✗ | **DeFund** ✗✗ |
|---|---|---|---|---|---|---|---|---|
| Worst-group | 47.2 | 64.4 | 77.2 | **81.1** | 10.2 | 1.1 | 17.1 | **77.9** |
| Average | **95.6** | 82.5 | 85.1 | 88.0 | 89.0 | 89.0 | 88.9 | **89.0** |

Table 11: (Semi-supervised learning) Worst-group and average accuracy evaluated on CelebA (blonde). Label fraction is set to 10%. Each first and second ✓marker represents whether the model requires information on target class or dataset bias in pretraining stage, respectively.

| Accuracy | LNL ✓✓ | EnD ✓✓ | JTT ✓✗ | CVaR DRO ✓✗ | ERM ✓✗ | SimCLR ✗✗ | **DeFund** ✗✗ |
|---|---|---|---|---|---|---|---|
| Worst-group (%) | 40.3 | 41.5 | 79.2 | 49.1 | 30.8 | 12.8 | **80.8** |
| Average (%) | **91.1** | 91.0 | 91.0 | 91.0 | 89.1 | 89.1 | 90.0 |

| Methods | Worst-group (%) | Average (%) |
|---|---|---|
| JTT | 70.6 | 86.6 |
| **DeFund** | **75.1** | **94.8** |

(a) Accuracy

|  |  | Male | |
|---|---|---|---|
|  |  | 0 | 1 |
| Blonde | 0 | 1558 | 53483 |
|  | 1 | 18417 | 1102 |

(b) Subsampled CelebA (blonde)

Table 12: (Semi-supervised learning) (**a**) Worst-group and average accuracy evaluated on subsampled CelebA (blonde). Label fraction is set to 10%. (**b**) Number of training samples for each group in subsampled CelebA (blonde). (Original dataset in Table 17d)

| | Conflict | Unbiased |
|---|---|---|
| SimSiam | 28.15 | 62.63 |
| + Rank reg | 23.40 | 59.65 |
| + Upweight | 56.12 | 65.44 |
| **DeFund**$_{Siam}$ | **60.37** | **67.78** |

(a) SimSiam

| | Conflict | Unbiased |
|---|---|---|
| VICReg | 32.33 | 64.58 |
| + Rank reg | 29.73 | 62.08 |
| + Upweight | 51.19 | 63.41 |
| **DeFund**$_{VIC}$ | **53.93** | **66.31** |

(b) VICReg

Table 13: Bias-conflict accuracy and unbiased accuracy evaluated on UTKFace (age). Last row corresponds to the full version of proposed framework which upweights misclassified samples identified by biased model. Results are averaged on 4 different random seeds. Accuracy is reported in (%).

Table 14: Comparison study on the depth of biased networks. Both networks are trained with target labels on CIFAR-10C (Bias ratio=95%). For UTKFace (age) and CelebA (makeup), both networks are pretrained with SimCLR followed by last linear layer training. Precision and recall are reported in (%).

| Networks | CIFAR-10C | | UTKFace (age) | | CelebA (makeup) | |
|---|---|---|---|---|---|---|
| | Precision | Recall | Precision | Recall | Precision | Recall |
| Shallow | 64.73 | **59.50** | 55.68 | 69.98 | 27.49 | **33.79** |
| ResNet-18 | **71.39** | 51.43 | **68.67** | **75.94** | **55.29** | 32.46 |

## C.4 SHALLOW NETWORK

Considering the inductive bias of neural networks towards encoding low effective rank representations in this paper, one may ask whether the shallow neural networks can easily learn such simple inductive bias and serve as a bias-capturing network. In this regard, we observe some pros and cons of using a shallow network as the bias model throughout experiments. Specifically, we use a simple convolutional network with three convolution layers as a counterpart of ResNet-18, with feature map dimensions of 64, 128 and 256, each followed by a ReLU activation and a batch normalization.

In the labeled setting, CIFAR-10C in Table 14 shows a tradeoff between precision and recall of the shallow network: The shallow network improves the recall of identified hard samples, i.e., the fraction of the bias-conflicting samples that are identified, because it is robust to the unintended memorization due to their fewer number of hyperparameters. However, it sacrifices the precision, i.e., the fraction of identified samples that are indeed bias-conflicting because its performance on the bias-aligned samples is degraded due to the low expressivity.

While the shallow network shows promising results with a simple dataset, the tradeoff worsens in the self-supervised setting with a larger dataset. Table 14 shows that the shallow network may suffer from bad precision. It is conventional wisdom that unsupervised learning benefits more from bigger models than its supervised counterpart (Chen et al., 2020). Considering this, the general performance of shallow networks may deteriorate in a large-scale self-supervised learning scenario. In this case, the identified error set $E$ contains too many false-positive bias-conflicting samples. While one may improve the performance with good care of hyperparameter tuning, e.g., depth of networks, learning rate, etc., it may be more laborious compared to the proposed framework, which has only a few scalar hyperparameters, e.g., $\lambda_{reg}$.

## C.5 HYPERPARAMETER ANALYSIS

While rank regularization biases the representations effectively, we do not argue that it is the optimal form of semantic bottleneck but rather that it highlights the unrecognized importance of controlling effective rank in encoding biased representations. In this regard, we examine the impacts of existing optimization hyperparameters on the effective rank and degree of bias in latent representations. Specifically, we investigated four candidates of bias controller through the lens of effective rank and generalizations: hardness concentration parameter $\beta$ of hard negative sampling (Robinson et al., 2020), temperature $\tau$ in InfoNCE (Oord et al., 2018) loss, strength of $\ell_2$ regularization $\lambda_{\ell_2}$ and the number of training epochs $T$.

**Hardness concentration parameter.** Recent works (Robinson et al., 2020; Cai et al., 2020; Tabassum et al., 2022) stress out the importance of negative examples that are difficult to distinguish from an anchor point. Several recent works propose algorithms on selecting informative negative samples, often controlled by hardness concentration parameter $\beta$ (Robinson et al., 2020) coupled with importance sampling. Robinson et al. (2021) conducted a synthetic simulation showing that increasing $\beta$ makes instance discrimination tasks more difficult, thereby enforcing the encoder to represent more complex features. Thus we aim to examine whether $\beta$ can contribute to learn a debiased representations with real-world dataset.

**Temperature.** A recent work on contrastive loss (Wang & Liu, 2021) have revealed that temperature $\tau$ can also control the strength of penalties on hard negative samples. Contrastive loss with high temperature turns out to be less sensitive to the hard negative samples (Robinson et al., 2020; 2021), thereby encouraging representations to be locally clustered while the uniformity of features on the hypersphere decreases (Wang & Isola, 2020). That being said, we hypothesized that the temperature $\tau$ may indirectly affect the effective dimensionality of representations, where large $\tau$ may decrease the effective rank.

$\ell_2$ **regularization and early-stopping.** Sagawa et al. (2019; 2020) underlines the importance of regularization for worst-case generalization where the naive upweighting strategy may fail if it is not coupled with strong regularization that prevents deep networks from memorizing upweighted bias-conflicting samples. In this regard, Liu et al. (2021) leverages capacity control techniques, e.g., strong $\ell_2$ regularization or early-stopping, to train complexity-constrained bias-capturing models. We investigate whether such regularizations can serve as a bias controller in self-supervised learning as well.

| Accuracy | 0.01 | 0.05 | 0.1 | 0.15 | 1 |
|---|---|---|---|---|---|
| Conflict | 35.8 | 36.3 | 37.5 | 37.6 | 36.6 |
| Unbiased | 65.6 | 65.6 | 66.6 | 66.5 | 66.0 |

|  | SimCLR | $\beta$=0.1 |
|---|---|---|
| Conflict | 62.0 | 64.2 |
| Unbiased | 78.9 | 80.7 |

(a) Biased linear evaluation

(b) Debiased linear evaluation

Table 15: Results of controlling concentration parameter $\beta$ on UTKFace (age). Accuracy is reported in (%). (**a**): Accuracy of linear evaluation without upweighting bias-conflicting samples. Each value in top row indicates $\beta$ used in pretraining. (**b**) Accuracy of linear evaluation with upweighting ground-truth bias-conflicting samples. Both models use $\lambda_{up} = 10$.

Table 16: Results of early-stopping on UTKFace. We denote $T$ as the number of training epochs.

| Attribute | Accuracy | $T = 5$ | $T = 10$ | $T = 15$ | $T = 20$ | $T = 25$ |
|---|---|---|---|---|---|---|
| Age | Bias-conflict (%) | 31.6 | 33.0 | 32.4 | 32.8 | 32.8 |
| | Unbiased (%) | 63.3 | 64.1 | 63.6 | 63.7 | 63.7 |
| Gender | Bias-conflict (%) | 54.6 | 54.0 | 53.5 | 53.4 | 54.5 |
| | Unbiased (%) | 72.1 | 72.0 | 71.8 | 72.2 | 72.7 |

**Results.** We evaluate each knob on generalizations with SimCLR. Table 16 and 15a show that impacts of both early-stopping and concentration parameter $\beta$ on generalizations are marginal, in contrast to the observations reported in supervised learning or synthetic simulations (Robinson et al., 2021). However, it still remains unclear whether the debiased representations can be encoded by controlling $\beta$. It is because the model may reach a biased solution even though it encodes debiased representations, if most samples in linear evaluation are bias-aligned, as discussed in the main paper. To preclude such confounding relationships, we conduct debiased linear evaluation with upweighting ground-truth bias-conflicting samples. Table 15a and 15b show that there was no significant difference in the performance gain of $\beta$ in biased and debiased linear evaluation, which implies that $\beta$ is not enough to fully debias representations.

Despite the failure of learning debiased representations with controlling $\beta$, biased representations can be learned by controlling temperature $\tau$, and strength of $\ell_2$ regularization in some cases. Figure 7a, 7b and 7c show that effective rank, temperature and bias-conflicting accuracy are highly correlated each other in both UTKFace and CelebA. It implies that the effective rank can serve as a metric of generalization performance and degree of bias in representations. While temperature control cannot be generalized to several non-contrastive learning methods (Chen & He, 2021; Bardes et al., 2021; Zbontar et al., 2021), this results imply that the temperature may serve as an effective bias controller for contrastive learning methods using InfoNCE loss. Moreover, stronger-than-typical $\ell_2$ regularization also limits the effective rank and bias-conflict accuracy to some extent in UTKFace (Figure 7d and 7e), while it fails to do so in CelebA.

This series of observations afford us a novel insight that both explicit (rank regularization) and implicit (temperature control, strong $\ell_2$ regularization) methods offer a way to train biased representations. However, it still remains unclear how to directly learn *debiased* representations. While increasing temperature or reducing effective rank bias representations, inverse does not always hold; Abnormally small temperatures cause the contrastive loss only focus on the nearest one or two samples, which heavily degenerates the performance (Wang & Liu, 2021). Moreover, we found that explicit decorrelation of feature components in SimCLR does not lead to debiased representations (not shown in figure).

To sum up, we provide useful recipes on learning biased representations, where rank regularization is mainly discussed in the main paper due to its intuitive insights, good performance and broad applicability. We hope these discussions facilitate in-depth studies about advanced algorithms on learning both biased and debiased representations in unsupervised manner.

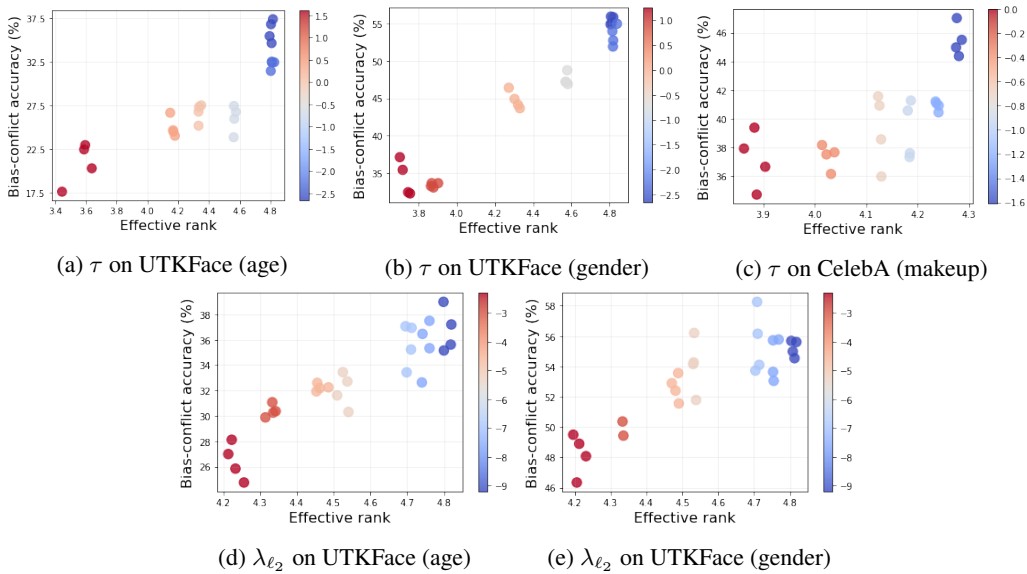

(a) $\tau$ on UTKFace (age)     (b) $\tau$ on UTKFace (gender)     (c) $\tau$ on CelebA (makeup)

(d) $\lambda_{\ell_2}$ on UTKFace (age)     (e) $\lambda_{\ell_2}$ on UTKFace (gender)

Figure 7: Analysis on temperature $\tau$ and strength of $\ell_2$ regularization $\lambda_{\ell_2}$. Effective rank and bias-conflict accuracy are measured with varying $\tau$ for (**a**, **b**, **c**), and $\lambda_{\ell_2}$ for (**d**, **e**). Standard deviation of bias-aligned accuracy on each experiment is $1.0\%$, $2.8\%$, $0.3\%$, $1.3\%$ and $1.7\%$ in order. Performance become quickly degenerated as $\lambda_{\ell_2}$ increases over $0.005$ in CelebA (makeup).

## D EXPERIMENTAL SETUP

### D.1 DATASETS

We mainly evaluate our debiasing framework on UTKFace (Zhang et al., 2017) and CelebA (Liu et al., 2015) in which several prior works has observed poor generalization performance due to spurious correlations. Example images are presented in Figure 8.

**UTKFace.** We first consider UTKFace dataset which is consist of human face images with varying `Race`, `Gender` and `Age` attributes. For each sensitive attribute, we categorize all samples into two groups. Specifically, for label associated with age, we assign 1 to samples with age $\leq 10$, and 0 to samples with age $\geq 20$ following (Hong & Yang, 2021). For label associated with race, we assign 1 to samples with race $\neq$ white, e.g., Black, Indian and Asian, and 0 to samples with race $=$ white. For label associated with gender, we assign 1 to female, and 0 to male. Based on this settings, we conduct binary classifications using (`Gender`, `Age`) and (`Race`, `Gender`) as (target, spurious) attribute pairs. Following Hong & Yang (2021), we construct a biased dataset by randomly truncating a portion of samples, where roughly $90\%$ of samples are bias-aligned in our setting. Pixel resolutions and batch size are $64 \times 64$ and 256, respectively.

**CelebA.** For CelebA, we consider (`HeavyMakeup`, `Male`) and (`Blonde Hair`, `Male`) as (target, spurious) attribute pairs, following (Nam et al., 2020; Hong & Yang, 2021; Sagawa et al., 2019). Pixel resolutions and batch size are $256 \times 256$ and 128, respectively. The exact number of samples for each prediction task is summarized in Table 17.

|   |   | A | |   |   | G | |   |   | M | |   |   | M | |
|---|---|------|------|---|---|------|------|---|---|-------|-------|---|---|-------|-------|
|   |   | 0 | 1 |   |   | 0 | 1 |   |   | 0 | 1 |   |   | 0 | 1 |
| G | 0 | 8229 | 822 | R | 0 | 4354 | 534 | H | 0 | 25789 | 54460 | B | 0 | 57214 | 53483 |
|   | 1 | 134 | 1346 |   | 1 | 435 | 5344 |   | 1 | 49804 | 163 |   | 1 | 18417 | 1102 |

(a) UTKFace (age)     (b) UTKFace (gender)     (c) CelebA (makeup)     (d) CelebA (blonde)

Table 17: Number of training samples for each prediction task. A for `Age`, G for `Gender`, R for `Race`, M for `Male`, H for `HeavyMakeup` and B for `Blonde Hair`.

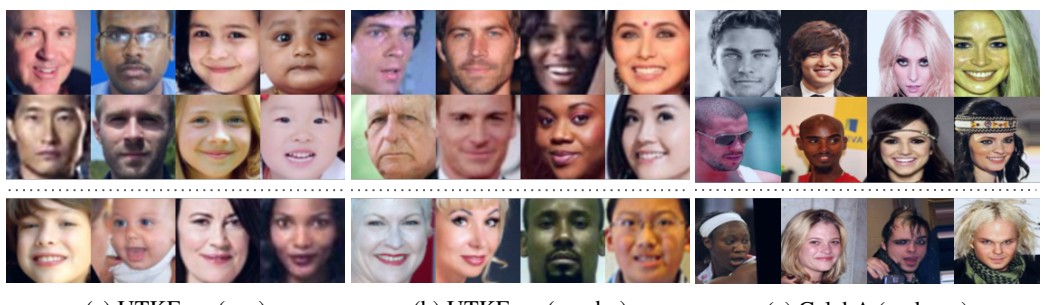

| (a) UTKFace (age) | (b) UTKFace (gender) | (c) CelebA (makeup) |

Figure 8: Example images of datasets. The images above the dotted line denote the bias-aligned samples, while the ones below the dotted line are the bias-conflicting samples. The group of two columns indicates each class of target attribute, i.e., `Gender, Race` and `HeavyMakeup`, respectively.

### D.2   RANK REDUCTION ANALYSIS

For CMNIST, we use a simple convolutional network with three convolution layers as a counterpart of ResNet-18, with feature map dimensions of 64, 128 and 256, each followed by a ReLU activation and a batch normalization. For CIFAR10-C and Waterbirds, we use ResNet-18 and ResNet-50 with pretrained weights provided in PyTorch torchvision implementations, respectively. The convolutional network is trained for 2000 iterations using SGD optimizer with inital learning rate 0.1 and decaying by 0.1 for every 600 iterations, following Zhang et al. (2021). For CIFAR10-C, ResNet-18 is trained for 10000 iterations using the Adam optimizer with learning rate 0.001. After training, misclassified training samples are identified as the bias-conflicting samples as in Table 1. For Waterbirds in Figure 2c, following the official implementation of JTT, ResNet-50 is trained for 300 epochs, and early-stopped with referring to the validation accuracy, using SGD optimizer with learning rate 0.0001.

### D.3   DEBIASING EXPERIMENTS

**Architecture details.** We use ResNet-18 back-bone architecture with pretrained weights provided in in PyTorch torchvision implementations. For projection networks in SimCLR, we use the MLP consists of one hidden layer with feature dimension of 512, followed by a ReLU activation. We employ a single linear classifier in downstream tasks for all self-supervised learning methods.

**Training details.** Both biased and main encoders are pretrained for 100 epochs on UTKFace, and 20 epochs on CelebA, by using Adam optimizer with learning rate of 0.0003. Cosine annealing scheduling (Loshchilov & Hutter, 2016) is leveraged with warmup for the first 20 epochs on UTK-Face, and 4 epochs for CelebA.

For biased encoders, we apply rank regularization with using $\lambda_{reg}$ of 0.3, 0.5, 0.01 and 0.03 for UTKFace (age), UTKFace (gender), CelebA (makeup) and CelebA (blonde), respectively. This values are selected by tuning $\lambda_{reg} \in \{0.0, 0.1, 0.3, 0.5, 1.0\}$ for UTKFace and $\lambda_{reg} \in \{0.0, 0.01, 0.02, 0.03, 0.05\}$ for CelebA. Specifically, we report the results of model with highest worst-group accuracy (for CelebA (blonde)), or bias-conflicting test accuracy over those with improved unbiased test accuracy compared to the SimCLR baseline. Same values are consistently used for upweighting in ablation study (Table 4). To emphasize the contribution of rank regularization, we do not control any other parameters, e.g., strength of $\ell_2$ regularization, temperature $\tau$, or number of training epochs. Specifically, we fix $\tau = 0.07$ and $\lambda_{\ell_2} = 0.0001$ for every experiment.

After pretraining, we conduct either linear evaluation or finetuning with using $\lambda_{up}$ of 10, 5, 8 and 15 for UTKFace (age), UTKFace (gender), CelebA (makeup) and CelebA (blonde), respectively. For UTKFace and CelebA (makeup), these values are selected by tuning $\lambda_{up} \in \{5, 8, 10\}$ using the above-mentioned decision rules, where $\lambda_{up} \in \{5, 8, 10, 15\}$ is compared for CelebA (blonde). Same values are consistently used in ablation study (Table 4). For linear evaluation, we train a linear classifier on top of pretrained main encoder for 3000 iterations on UTKFace, and 5000 iterations on CelebA, with using learning rate of 0.0003 and upweighting identified bias-conflicting samples.

For semi-supervised learning, we finetune the whole main model for 5000 iterations, with using SGD optimizer, momentum of 0.9, $\lambda_{\ell_2} = 0.1$, learning rate of 0.0001, and $\lambda_{up} = 8, 15$ for CelebA (makeup) and CelebA (blonde), respectively.

**Data augmentations.** Following SimCLR, we generate multiviewed batch with random augmentations of (a) random resized crop with setting the scale from 0.2 to 1, (b) random horizontal flip with the probability of 0.5, (c) random color jitter (change in brightness, contrast, and saturation) with the probability of 0.8 and scale of 0.4, (d) random gray scaling with the probability of 0.2. In linear evaluation and finetuning, we only apply random horizontal flip. Same augmentation pipeline is applied to both SimSiam and VICReg.

**Baselines.** For a fair comparison, we tune hyperparameters of other baselines using the same ResNet-18 back-bone architecture. We use the official implementation of JTT which also includes that of CVaR DRO. Other baselines are reproduced by ourselves with referring to original papers. LNL is trained for 20 epochs on UTKFace, and 40 epochs on CelebA, with using Adam optimizer and learning rate of 0.001. For EnD, we set the multipliers $\alpha$ for disentangling and $\beta$ for entangling to 1. For JTT, we tune the upweighting factor $\lambda_{up} \in \{20, 50, 80\}$ and number of training epochs $T \in \{30, 40, 50\}$, following the original paper. For CVaR DRO, we tune the size of the worst-case subpopulation $\alpha \in \{0.1, 0.2, 0.5\}$. For SimSiam and VICReg, the architectures for the additional layers followed the official implemenation of each method, where the hyperparameters for the training is identical to the SimCLR case. For C.3, $\lambda_{reg} = 0.001$ for **DeFund**$_{\text{Siam}}$ and $\lambda_{reg} = 0.1$ for **DeFund**$_{\text{VIC}}$.

