# OpenReview forum: "Self-supervised debiasing using low rank regularization"
_ICLR.cc/2023/Conference — Submitted to ICLR 2023_

### Official Review · Reviewer_3q4H · 2022-10-23

**Confidence:** 4
**Correctness:** 2
**Technical Novelty And Significance:** 2
**Empirical Novelty And Significance:** 3
**Recommendation:** 3

**Clarity, Quality, Novelty And Reproducibility:**

Questions and Remarks:
1. Which network has been used to perform the analyses in Section 2.2? Do these analyses hold also for other architectures, e.g. ViT?
2. Are you plotting the effective rank of the representations in the network?
3. In Table 1 and other tables, how do you still the error set E or misclassified training examples for the baselines? How do you do it for your method?
4. It would be good to compare to Deep Feature Reweighting, given that they also retrain the last linear layer to robustify the model

**Strength And Weaknesses:**

Strengths:
- *The paper studies an important problem*.
- *Novel perspective on an existing problem*: it deserves to be stated more clearly in the paper, but the idea of exploiting unlabeled data to find "easy-to-learn" features to robustify models in the labeled setting is interesting. Also, the evidence towards establishing a correlation between effective rank and the degree of bias in the dataset is interesting and make sense.

Weaknesses:
- *Lack of motivation and clarity of exposition*: I found the paper hard to follow. It is quite dense and contributions are hard to parse at the beginning. Let me try to summarize the two contributions. The first is to propose a rank-suppression mechanism to find biased representations. The second is the will to exploit un-labeled data to mine more plausible features that could act as spurious correlations for the task, which could help for example in the absence of a lot of labeled data. I find that these two contributions can be more clearly stated along with the motivations supporting these contributions.

- *A conceptual weakness does not seem to be discussed*: spurious correlations are defined with respect to the label by definition. The paper claims that rank regularization exacerbates "short-cuts" or "easy-to-learn" features. It is reasonable to think that spurious and easy-to-learn feature match in the labeled setting, but the connection between the two might be more tricky in the unlabeled setting. Is a model capturing an easy-to-learn feature lead to always better performance against spurious correlations? I could think of a synthetic dataset in which an easy-to-learn feature is perfectly un-correlated with the labels? Would your regularization scheme just learn that feature thus making classification and thus bias-conflicting examples meaningless?

- *Findings lack support*: Another main weakness of the paper is that the contributions are a bit "dispersed":
a) With respect to the first contribution, I would expect more extensive analysis of how rank-regularization compare (in the labeled setting) to other methods of finding bias-conflicting examples e.g. JTT, and Forgettables examples (Yaghoobzadeh et. al, 2021, "Increasing robustness to spurious correlations..."). Right now the authors compare with ERM in Table 1, without giving much details on the experimental setting. Then, the authors could go on in justifying why rank-regularization can be used in the unlabeled setting (differently from other approaches that require labels).
b) When talking about un-labeled data, it is natural to ask whether standard pre-trained self-supervised/supervised models can be used in this setting. In the paper, the authors limit themselves to the setting where the encoder is trained only on *unlabeled* data for the task, but what if we are starting from a pre-trained model (classical setting of JTT / Deep Feature Reweighting)? How can the approach be applied?
c) The authors test their model in CelebA and UTKFace. Waterbirds and CelebA are classical settings for this problem, and the classical CelebA split is (hair, gender). Why do the authors skipped these settings?

- *Weak results in Table 2*: In Table 2, the scores for DeFund are bolded but are less than ERM on some columns and JTT on most of the columns. I understand the model is not using Y in the pre-training phase, but the question one could ask is *why not* given that it is using those labels in the fine-tuning stage and thus the exact same information is used to report those numbers?


**Summary Of The Paper:**

NNs are generally victim of spurious features which are strongly correlated with the label in the training set, but might not be robust for unseen samples. This paper proposes a method to reduce the reliance on spurious correlations.

The strategy is to first find examples that are bias-conflicting with a *bias model* and then train a robust, *main model* by up-weighting those examples when learning the task classifier. The bias model is trained with a rank-regularization loss which shrinks the effective rank of the the representations in the NN: a small effective rank describes a paucity of features that the network capture, and these likely correspond to easy-to-learn features in a spurious correlation setting. The authors test their approach in the unlabeled setting, using SimCLR/SiamSiam/VicREG models, regularized with their rank loss for obtaining the bias model.

Summarizing the contributions, the paper mainly argues two novelties: 1) rank regularization helps in finding biased examples; 2) one can apply rank regularization also to the unlabeled setting, so that one can exploit unlabeled data for a task to prevent spurious correlations.

**Summary Of The Review:**

The paper addresses an important problem. The study on the effective rank is interesting and possibly impactful. The unlabeled perspective should be discussed in more details: implications and applicability to standard pre-trained models should be discussed. Overall, the paper feels a bit dispersed at the moment: I am not sure I can confidently state that a particular technique works better than the baselines or take away a clear message from this paper.

---

> ### Author Response · Authors · 2022-11-16
> **Official Comment by Paper3840 Authors [1/4]**
>
> We appreciate your insightful and constructive comments that help us to improve our paper. Here we have tried our best to fully clarify all the issues in the point-by-point responses (the reviewer’s comment is highlighted, followed by our reply).
>
> > **Q1.** Lack of motivation and clarity of exposition
>
> Thank you for the constructive feedback. Per your request, we have clarified the motivation/contribution in the introduction section by adding a summary of our contributions.
>
> We would like to respectfully note that the motivation of our work, as other reviewers recognized, is to train a debiased classifier by fully exploiting unlabeled samples lacking both bias and target labels. We would like to kindly remind the reviewer that in the practical situation, we often encounter a potentially biased large-scale dataset that is mostly composed of unlabeled samples. Unfortunately, several limitations may exist against collecting more labeled samples, for example,  labeling budget, expert-level knowledge required for labeling, data privacy, etc. However, most existing supervised debiasing frameworks heavily depend on labeled data, which raises concerns about its robustness with scarcely labeled samples. Accordingly, our work tried to address the following open problems/questions:
>
> - What is the fundamental difference between biased and debiased representations?
> - Can we learn biased/debiased representations by using unlabeled samples?
> - Is supervised debiasing robust with the small number of labeled samples?
> - How can bias-conflicting samples be discovered by leveraging information from unlabeled samples?
> - Many recent works have reported the limitations of self-supervised learning (SSL) in OOD generalization. How can we overcome such limitations?
>
> To this end, we have successfully demonstrated a label-free regularization in terms of rank by leveraging fully unlabeled samples.
>
> > **Q2.** A conceptual weakness does not seem to be discussed: The paper claims that rank regularization exacerbates "short-cuts" or "easy-to-learn" features. It is reasonable to think that spurious and easy-to-learn feature match in the labeled setting, but the connection between the two might be more tricky in the unlabeled setting. Is a model capturing an easy-to-learn feature lead to always better performance against spurious correlations?
>
> Thank you for the insightful comments. As mentioned in the first paragraph of the introduction, we follow conventions of several recent works which explain spurious correlations through the lens of simplicity bias or the “Principle of Least Effort” [1, 2]. Specifically, recent works [3, 4] have revealed that neural networks learn to rely on the spurious correlation only when it is “easier” to learn than the desired knowledge. Accordingly, if we encounter a biased classifier or dataset, it is reasonable to assume that the unknown easy-to-learn shortcut may reside in the dataset, as discussed in several previous works [5, 6].
>
> The reviewer is also kindly reminded that we do not argue that all easy-to-learn features are spuriously correlated. We agree that some innocent easy-to-learn features may be perfectly uncorrelated with the labels, and every easy-to-learn feature does not necessarily have to be spurious shortcuts. However, in the presence of some spurious shortcuts, such shortcuts may be easier-to-learn than the desired but complex/high-level relationships among features, as reported by several works on learning visual representations. Removing such shortcuts would help the networks to learn the desired characteristics successfully.  To preclude the bias-capturing models from learning complex intended features, the proposed rank regularization controls the semantic diversity. As a result, hard-to-learn invariant features are ruled out, while easy-to-learn shortcuts survive in biased representations and eventually dominate the downstream classification.
>
> In the above context, we would like to answer your question:
>
> > **Q:** Does capturing an easy-to-learn feature always lead to better performance against spurious correlations?
>
> **A:** Considering spurious correlations, the model mainly capturing easy-to-learn features would severely suffer from poor OOD generalization.  In fact, if the desired feature were to easier to learn, we would have not been concerned about the bias at the first place.

---

> > ### Author Response · Authors · 2022-11-16
> > **Official Comment by Paper3840 Authors [2/4]**
> >
> > > **Q3.** Comparisons to other algorithms mining bias-conflicting samples in a supervised setting. Right now the authors compare with ERM in Table 1, without giving much details on the experimental setting.
> >
> > Per your request, we have added experimental details in appendix C. experimental setup section. We also found and fixed typos on the bias ratio in Table 1.
> >
> > Specifically, following the official implementation of JTT, the precision and recall of identified bias-conflicting samples are compared in Resp_Table 6 below. $\lambda_{\ell_2}=1$ is used for JTT following the official command, while $\lambda_{\ell_2}=1e-4$ is used for ours to focus on the contribution of rank regularization. The number of training epochs is selected by tuning $T \in \{20, 40, 60\}$ for both JTT ($T=60$) and ours ($T=40$). As reported, the proposed rank regularization improves the precision and recall of identified bias-conflicting samples by using sufficiently large $\lambda_{reg}$. Even though we agree that rank regularization may be a viable option for the minority mining algorithms in a supervised learning setting, we defer extensive discussions about this possibility to focus on the unsupervised learning settings.
> >
> > **Resp_Table 6.** Precision and recall of identified bias-conflicting samples. All the networks are trained on Waterbirds dataset by using ResNet-50 as a base architecture. Detailed experimental setup is provided in the appendix.
> > | Metrics       | ERM   | JTT   | Ours ($\lambda_{reg}$=5) | Ours ($\lambda_{reg}$=10) |
> > |---------------|-------|-------|--------------------------|---------------------------|
> > | Precision (%) | 37.84 | 48.95 | 48.91                    | **54.77**                 |
> > | Recall (%)    | 11.67 | 48.75 | 46.67                    | **55.00**                 |
> >
> > We agree that our work is relevant to the suggested works, and thank the Reviewer for providing references. The revised appendix contains more discussions on the related work including the suggested ones. Yaghoobzadeh et al. shows that forgettables contain more minority examples, and proposes a novel robust learning framework by fully exploiting the identified forgettable examples. We believe that these results may facilitate valuable discussions on the relationship between careful data selection strategy and OOD generalization. However, these methods inevitably rely on labels to some extent. In contrast, the proposed rank regularizer in Eq.(3) does not require any labels on the bias or target attributes. Instead, it fully exploits the information from the relationship between feature components that most debiasing algorithms have overlooked.
> >
> > > **Q4.** The authors test their model in CelebA and UTKFace. Waterbirds and CelebA are classical settings for this problem, and the classical CelebA split is (hair, gender). Why do the authors skipped these settings?
> >
> > We would like to assure the reviewer that the dataset in Figure 2 (CMNIST, CIFAR-10C, Waterbirds) was not included in section 4 as the encoder trained with SimCLR already encodes invariant representations w.r.t simple spurious attributes, as mentioned in section 4.1. Accordingly, existing debiasing frameworks, including ours, are intended to deal with more challenging situations where the standard models fail to generalize.
> >
> > That said, we also conducted additional linear evaluation and semi-supervised learning tasks on CelebA (blonde). Section C.2 in the revised paper shows that DeFund outperforms not only every self-supervised baseline, but also ERM, CVaR DRO, and LfF in linear evaluation. Moreover, it shows that DeFund outperforms all the other baseline methods in a semi-supervised setting, which is consistent with Table 3 of the main paper.
> >
> > > **Q5.** Weak results in Table 2: why not given that it is using those labels in the fine-tuning stage and thus the exact same information is used to report those numbers?
> >
> > To further clarify the experimental setting of linear evaluation to avoid potential confusion,  we have highlighted the experimental setup of linear evaluation in the revised paper: “First, following the conventional protocol of self-supervised learning, we conduct linear evaluation (...) by using ~~every labeled~~ → using target labels of every training sample”. Our pseudocode may further clarify the difference between linear evaluation and semi-supervised learning protocol.
> >
> > As mentioned in section 3, linear evaluation is mainly opted for evaluating self-supervised learning methods, which do not leverage label-associated information at all during learning representations. With this regard, we argue that a more fair comparison can be made in semi-supervised learning, where labeled samples are provided equally to both supervised and self-supervised methods. While semi-supervised learning is a more practical scenario, we also report linear evaluation results to compare the quality of biased representations and identified bias-conflicting samples in controlled settings.

---

> > > ### Author Response · Authors · 2022-11-16
> > > **Official Comment by Paper3840 Authors [3/4]**
> > >
> > > > **Q6.** When talking about un-labeled data, it is natural to ask whether standard pre-trained self-supervised/supervised models can be used in this setting. In the paper, the authors limit themselves to the setting where the encoder is trained only on unlabeled data for the task, but what if we are starting from a pre-trained model (classical setting of JTT / Deep Feature Reweighting)? How can the approach be applied?
> > >
> > > We would like to respectfully remark that using a pretrained self-supervised/supervised model, although it is a viable option, may limit its potentially wide applicability of debiasing methods.  Accordingly, our study aims to overcome fundamental limitations in more challenging self-supervised debiasing problems. To make our point more clear,  the reviewer is kindly reminded that  pretrained classifiers can be categorized as follow:
> > >
> > > 1. Supervised model (pretrained on labeled samples of interest): Several existing supervised debiasing methods leverage pretrained classifiers to identify bias-conflicting samples (JTT), or probe invariant features for the reweighting strategy (Deep Feature Reweighting; DFR). However, these methods pretrain the model from scratch using a fully labeled dataset or assume the existence of such pretrained ERM models, where both cases are infeasible when most training samples are fully unlabeled. For example, JTT pretrains the network by controlling its capacity, e.g., strong l2 regularization, which inevitably requires a labeled dataset. While DFR is compatible with existing pretrained networks, it still remains unclear how to obtain such pretrained networks with limited access to the training labels. Please refer to Q7 for more discussions on DFR.
> > >
> > > 2. Supervised model (pretrained on a large, diverse, and potentially relevant dataset): One may try transfer learning to adapt a so-called source model pretrained on a large dataset, e.g., ImageNet, to the task of interest, by finetuning with a small number of labeled samples included in their unlabeled dataset. While our ERM baselines indeed use ImageNet-pretrained weights provided in PyTorch torchvision implementations, it was insufficient to robustify networks. Moreover, a recent study [7] has revealed the potential pitfalls of transfer learning: bias transfer, or the tendency for biases of the source model to persist even after adapting the model to the target class. It highlights the potential limitations of using pretrained models for robust learning.
> > >
> > > 3. Self-supervised model (pretrained on a whole dataset of interest): As mentioned in the introduction, recent works on self-supervised learning have reported that self-supervised learning may still suffer from poor OOD generalization when such dataset bias remains after applying data augmentations. These failure modes indeed motivate us to design a new framework to overcome such limitations and study the property of biased representations.
> > >
> > > > **Q7.** It would be good to compare to Deep Feature Reweighting, given that they also retrain the last linear layer to robustify the model.
> > >
> > > Per your request, we compare the proposed framework with DFR below:
> > >
> > > - While both DFR and the proposed framework can mitigate the bias in representations by retraining the last linear layer, our method is not restricted to such last-layer retraining. For example, in a semi-supervised learning scenario, we can fine-tune representations by fully exploiting both unlabeled and labeled samples, which improves the performance in Table 3. In contrast, DFR trains a linear classifier while freezing the pretrained representations as-is.
> > >
> > > - More importantly, DFR requires the pretrained networks or fully labeled datasets whereas we consider a more challenging scenario without such assumptions.
> > >
> > > - Moreover, DFR does not mine bias-conflicting samples in the training set. Specifically, DFR trains a new classification head from scratch on the available group-balanced data $\hat{D}$. In DFR, the reweighting dataset $\hat{D}$ consists of a group-balanced subset of the training or validation data. In other words, DFR is not designed to identify the bias-conflicting samples but exploits the existing group annotations. Considering practical situations with several limitations against collecting more labeled samples, it remains unclear how to obtain a group-balanced dataset with a sufficient number of samples in the absence of prior information on the dataset bias. In contrast, the proposed framework can leverage the explicit set $\hat{D}$ if accessible, as well as identify the bias-conflicting samples in the training set.

---

> > > > ### Author Response · Authors · 2022-11-16
> > > > **Official Comment by Paper3840 Authors [4/4]**
> > > >
> > > > > **Q8.** Which network has been used to perform the analyses in Section 2.2? Do these analyses hold also for other architectures, e.g. ViT?
> > > >
> > > > To clarify the experimental setting of analyses in Section 2.2, we would like to remind the reviewer that we used a simple convolutional network with three hidden-layers for CMNIST and ResNet-18 for CIFAR-10C with pretrained weights provided in PyTorch torchvision implementations. More implementation details are added in the appendix of the revised paper. We observed similar rank reduction trends in ViT, which are also provided in the appendix.
> > > >
> > > > > **Q9.** Are you plotting the effective rank of the representations in the network?
> > > >
> > > > We measured the effective rank of representations in the penultimate layer of the network or the output representations of the encoder, as mentioned in sections 2.2, page 3, and 2.3, page 4. We added explanations in the caption of fig. 1.
> > > >
> > > > > **Q10.** In Table 1 and other tables, how do you still the error set E or misclassified training examples for the baselines? How do you do it for your method?
> > > >
> > > > In Table 1, ERM and the rank-regularized models are trained with the same iterations, learning rate, optimizer, etc. The only difference is whether to use rank regularization or not. The error set E is distilled after training by evaluating training samples, likewise for the other baselines mining minorities (JTT // SimCLR in ablation study Table 4). Please refer to appendix section C.2, Training details paragraph for more information.
> > > >
> > > > _______________________________________________
> > > > ### References.
> > > >
> > > > [1] Geirhos, Robert, et al. "Shortcut learning in deep neural networks." Nature Machine Intelligence (2020)
> > > >
> > > > [2] Shah, Harshay, et al. "The pitfalls of simplicity bias in neural networks.", NIPS 2020
> > > >
> > > > [3] Nam, Junhyun, et al. "Learning from failure: De-biasing classifier from biased classifier." NIPS 2020
> > > >
> > > > [4] Lee, Jungsoo, et al. "Learning debiased representation via disentangled feature augmentation." NIPS 2021
> > > >
> > > > [5] DeGrave, Alex J., et al. "AI for radiographic COVID-19 detection selects shortcuts over signal." Nature Machine Intelligence (2021)
> > > >
> > > > [6] McCoy, R. Thomas, Ellie Pavlick, and Tal Linzen. "Right for the wrong reasons: Diagnosing syntactic heuristics in natural language inference." ACL 2019
> > > >
> > > > [7] Salman, Hadi, et al. "When does Bias Transfer in Transfer Learning?." (2022)

---

### Official Review · Reviewer_cTbQ · 2022-10-24

**Confidence:** 2
**Correctness:** 3
**Technical Novelty And Significance:** 3
**Empirical Novelty And Significance:** 3
**Recommendation:** 6

**Clarity, Quality, Novelty And Reproducibility:**

- The paper is well written and organized. The code is also attached in the supplementary.

**Strength And Weaknesses:**

**Strength:**

- Well motivated and intersting idea. The experiements and ablation studies are designed carefully.

- The improvemnts are significant observed from Linear evaluation and Semi-supervised learning experiments.

**Weaknesses:**

- It would be better to report error bars to show the stableness of the result.

- If this two-stage training framework would increase the training time compared to other approaches?

**Summary Of The Paper:**

This paper proposes a debiasing framework using low-rank regularization and self-supervised learning techinque. Specifically, Authors find that rank regularization may force deep networks to focus on spurious attributes and the biased models with strong regularization can effectively probe out the bias-conflicting samples in the training set which is robust to the unintended memorization of bias-conflicting samples. Inspired by the obervation, authors desings two-stage training framework to debias the model. Experiments are verified on UTKface (age), UTKFace (gender) and CelebA to verify its effectiveness.

**Summary Of The Review:**

- I am not very familiar with this direction, along with the related work. From my perspective, I think this paper has some intersting points. Thus I give the score of 6 at the current time and will make my final decision aftering reading other reviews.

---

> ### Author Response · Authors · 2022-11-16
> **Official Comment by Paper3840 Authors**
>
> We appreciate your insightful and constructive comments that help us to improve our paper. Here we have tried our best to fully clarify all the issues in the point-by-point responses (the reviewer’s comment is highlighted, followed by our reply).
>
> > **Q1.** It would be better to report error bars to show the stableness of the result.
>
> Thank you for the constructive feedback. We have added standard deviations in the main results of the revised paper.
>
> > **Q2.** If this two-stage training framework would increase the training time compared to other approaches?
>
> While the proposed framework comprises two stages, we would like to gently note that the second retraining/finetuning stage costs affordable training time. In linear evaluation, we only train the linear classifier with a few epochs, e.g., about 30 epochs for UTKFace, and 4 epochs for CelebA, likewise as semi-supervised learning. Thus the additional computational overhead is negligible compared to standard self-supervised learning.

---

### Official Review · Reviewer_datJ · 2022-10-24

**Confidence:** 3
**Correctness:** 3
**Technical Novelty And Significance:** 3
**Empirical Novelty And Significance:** 2
**Recommendation:** 6

**Clarity, Quality, Novelty And Reproducibility:**

Clarity: The paper has a good clarification.

Quality: I checked the appendix. The paper has good writing.

Novelty: The paper has its novelty as far as I know.

Reproducibility: I believe the experiments part can be reproduced.


**Strength And Weaknesses:**

Strength:
1. The paper has a clear motivation and insights. The authors first find an interesting observation that the bias is a low-rank representation, and then provide a trivial but effective solution.
2. The proposed methods have a good performance on some complex datasets, e.g. CelebA.

Weakness and Questions:
1. This is pure empirical work. There are some claims and statements that may need analysis support or more empirical support, e.g. (1) Why are Equations (2) and (3) a good way to estimate the effective rank? Why not use the nuclear norm, which is a convex relaxation of a matrix rank? (2) Why does the low-rank regularization guarantee learning more spurious correlations features rather than intrinsic features? If we only constrain the rank, how should we know which feature it tends to learn? Figure 1d is not enough to answer this question from my perspective. I need more evidence empirically (e.g. feature visualization for the bias model) or theoretically.
2. In synthetic simulation experiments under Section2, the bias correlation is high, (larger than 95%). What will happen in a more natural setting, e.g. 80%? In Table 1, what is the meaning of Bias ratio =1%? Is it the same as the Bias ratio = 99%?
3. In Figure 1f, the two curves for CIFAR10-c have marginal differences.
4. From my perspective, the critical part of the method is to find bias-conflicting samples, or we may call them hard samples. I wonder whether low-rank regularization is the only solution. One potential idea is to use a shallow network (e.g. two-layer neural network) as the bias model rather than using low-rank regularization. I am curious about whether the shallow network method work, considering that it may just learn simple inductive bias as the authors claim.


**Summary Of The Paper:**

The paper studies debiasing spurious correlations and finds the inductive bias towards encoding low effective rank representations. The paper proposes a low-rank regularization method to debias. First, one bias model learns representations with low-rank regularization to amplify the bias. Then, we use the bias model to find the biased conflicting samples. Finally, use larger weights on biased conflicting samples in training the main models. Their proposed methods do not require information on target class or dataset bias information. The authors evaluate their methods on UTKFace and CelebA, and the results outperform most state-of-the-art.

**Summary Of The Review:**

The paper has its novelty, while some questions I mentioned in the Weakness part blocked me. I may adjust my rating based on the rebuttal.

---

> ### Author Response · Authors · 2022-11-16
> **Official Comment by Paper3840 Authors [1/3]**
>
> We appreciate your insightful and constructive comments that help us to improve our paper. Here we have tried our best to fully clarify all the issues in the point-by-point responses (the reviewer’s comment is highlighted, followed by our reply).
>
> >**Q1.** Why are Equations (2) and (3) a good way to estimate the effective rank? Why not use the nuclear norm, which is a convex relaxation of a matrix rank?
>
> Thank you for the insightful feedback. We design a rank regularizer motivated by the observations in Figure 1a, b, which indicate that the features may become highly correlated in biased settings. Moreover, we consider some practical aspects that Eq.(3) is widely used in self-supervised learning frameworks [1, 2]; it is easy to implement and may provide a new perspective on the conventional loss term to self-supervised learning practitioners.
> In this regard, we agree that a better measure may exist to implement rank regularization in practice. From our preliminary analysis in Resp_Table 2 below, the performance of nuclear norm regularization was not significantly different from ours if one carefully chooses the degree of regularization.
>
> **Resp_Table 2.** Comparison study of the nuclear norm and effective rank regularization. Both regularized models are trained on CIFAR-10C (Bias ratio=95%). $\lambda_{reg}=50$ and $\lambda_{reg}=0.5*\frac{1}{512}$ are used for effective rank and nuclear norm regularization, respectively, where $512$ corresponds to the dimension of representations.
> | Methods         | Align (%) | Conflict (%) | Unbiased (%) |
> |---------------------|-----------|--------------|--------------|
> | Nuclear norm reg.   | 96.00     | 23.53        | 30.78        |
> | Effective rank reg. | 96.50     | 23.77        | 31.04        |
>
> That said, for the case of the nuclear norm, top singular values are significantly large, as shown in Figure 1f, so that the distributional property of singular values may be obfuscated in the nuclear norm as shown in Resp_Table 3. This suggests that while nuclear norm may be a strong candidate for rank regularizer with a solid theoretical background, we recommend using the effective rank in feature analysis.
>
> **Resp_Table 3.** Normalized nuclear norm (norm / dimension) measured in CMNIST and CIFAR-10C with varying bias ratios.
> | Dataset   | Unbiased | 95%  | 98%  | 99%  | 99.5% |
> |-----------|----------|------|------|------|-------|
> | CMNIST    | 2.47     | 2.56 | 2.56 | 2.59 | 2.46  |
> | CIFAR-10C | 7.12     | 5.92 | 6.34 | 6.54 | 6.51  |
>
>
> > **Q2.** In synthetic simulation experiments under Section2, the bias correlation is high, (larger than 95%). What will happen in a more natural setting, e.g. 80%? In Table 1, what is the meaning of Bias ratio =1%? Is it the same as the Bias ratio = 99%?
>
> We apologize for the typo mistake as the bias ratio should be 99% in Table 1. We set the bias ratio in section 2 following the conventions in related works [5, 6]. The unbiased accuracy decreases regularly following the bias ratio, as shown in Table 7 of the appendix. For example, by fitting linear regression with effective rank (y) in fig. 1c and unbiased test accuracy (X) of CMNIST in Table 7, one can obtain y=0.0106*X + 3.1736, with p-value < 0.005 for the slope coefficient.
>
> We apply the rank regularization to the moderately biased CMNIST, e.g., bias ratio = 60%. Table 9 in the appendix shows that rank regularization works well in a natural setting. Moreover, Waterbirds in Figure 2c, UTKFace and CelebA correspond to a more general dataset with higher resolution and a moderate bias ratio. These observations suggest that the effective rank, bias ratio, and unbiased accuracy are closely related.
>
> > **Q3.** In Figure 1f, the two curves for CIFAR10-c have marginal differences.
>
> We would like to respectfully note that the difference between biased and debiased representations resides in the remaining majority of singular values except the top few values. Specifically, the top few normalized singular values of biased representations are similar to that of unbiased representations, while the remaining values decay significantly faster in biased representations. For example, there is about a 35.7% decrease in CMNIST, and 38.4% decrease in CIFAR10-C for the 40-th normalized singular value comparing the biased one to the unbiased one.

---

> > ### Author Response · Authors · 2022-11-16
> > **Official Comment by Paper3840 Authors [2/3]**
> >
> > > **Q4.** Why does the low-rank regularization guarantee learning more spurious correlations features rather than intrinsic features? If we only constrain the rank, how should we know which feature it tends to learn? Figure 1d is not enough to answer this question from my perspective. I need more evidence empirically (e.g. feature visualization for the bias model) or theoretically.
> >
> > Thank you for the insightful comments. To understand the relationship between rank regularization and spurious correlations more deeply, we visualize the pretrained representations with varying degrees of bias. We first trained deep networks on: (a) unbiased CMNIST, (b) biased CMNIST (bias ratio=95%) without rank regularization and (c) with rank regularization (\lambda=50). Then, we train the auxiliary decoder, which reconstructs the bias-conflicting images from freezed latent representations of each pretrained network.
> >
> > Figure 3 in appendix C.1 shows that the reconstructed results are evidently different for each case. First, the decoder successfully reconstructs the foreground digit from the (a) unbiased representations, while the background color is completely changed in some cases. It implies that unbiased representations may lack information on spuriously correlated attributes, i.e., background color. However, both digit and color are well reconstructed in (b) biased case, implying that the biased model encodes both spurious and invariant features. Intriguingly, the decoder fails to reconstruct bias-conflicting images from the (c) biased representations pretrained with rank regularization. Specifically, the foreground digit is blurred, and its class is changed following the background color.
> >
> > Based on these observations, we summarize some key insights: First, the rank-regularized representation may lose its information on harder-to-learn invariant features. While the reconstructed images in (a) or (b) preserve the detailed class, shape, and style of the foreground digit, such properties are deteriorated in (c), implying the loss of feature discriminability and informative signals. Second, with limited semantic diversity, the rank-regularized model fails to identify the true underlying independent generative factors for multidimensional data; it may rather encode feature components entangled with both spurious and invariant attributes. In other words, the proposed low-rank regularization prevents features from encoding discriminative information independently. It is indirectly reflected in (c) that the digit class of the reconstructed image is erroneously determined by the spurious attribute, i.e., background color.
> >
> > The experiments showed that our rank regularization encourages the network to focus more on the spurious correlations in a way that minimizes semantic diversity. We hope that the evidence in this paper may provide clues for the important theoretical findings on the relationship between spurious correlations and deep representations. Please note that our findings are in line with the simplicity bias [3,4] of neural networks: While the biased models are known to prefer simple solutions (low-rank in our special case) with perfect-fit on the training samples, the proposed rank-regularized network prevents unintended memorization of minorities and then encourages models to learn simple solutions with best-fit under the regularization, i.e., focusing on the majority samples.

---

> > > ### Author Response · Authors · 2022-11-16
> > > **Official Comment by Paper3840 Authors [3/3]**
> > >
> > > > **Q5.** From my perspective, the critical part of the method is to find bias-conflicting samples, or we may call them hard samples. I wonder whether low-rank regularization is the only solution. One potential idea is to use a shallow network (e.g. two-layer neural network) as the bias model rather than using low-rank regularization.
> > >
> > > Thank you for the insightful comments. Per your request, we have conducted additional experiments and observed some pros and cons of using a shallow network as a bias model. Specifically, a three-layer convolutional network is implemented as our shallow network as a counterpart of ResNet-18. In the labeled setting, Resp_Table 4 shows a tradeoff between precision and recall of the shallow network: The shallow network improves the recall of identified hard samples, i.e., the fraction of the bias-conflicting samples that are identified, because it is robust to the unintended memorization due to their fewer number of hyperparameters. However, it sacrifices the precision, i.e., the fraction of identified samples that are indeed bias-conflicting because its performance on the bias-aligned samples is degraded due to the low expressivity.
> > >
> > > **Resp_Table 4.** Precision and recall of the identified bias-conflicting samples by shallow networks and rank-regularized ResNet-18. Both networks are trained on CIFAR-10C (bias ratio=95%).
> > > | Networks  | Precision (%) | Recall (%) |
> > > |-----------|---------------|------------|
> > > | Shallow   | 64.73         | **59.5**       |
> > > | Ours | **71.39**         | 51.43      |
> > >
> > > While the shallow network shows promising results with a simple dataset, the tradeoff becomes worsens in the self-supervised setting with a larger dataset. Resp_Table 5 below shows that the shallow network may suffer from bad precision. It is conventional wisdom that unsupervised learning benefits more from bigger models than its supervised counterpart [7]. Having this in mind, the general performance of shallow networks may deteriorate in a large-scale self-supervised learning scenario. In this case, the identified error set $E$ contains too many false-positive bias-conflicting samples. While one may improve the performance with good care of hyperparameter tuning, e.g., depth of networks, learning rate, etc., it may be more laborious compared to the proposed framework, which has only a few hyperparameters, e.g., $\lambda_{reg}$. Detailed simulation settings and discussions are provided in the appendix of the revised paper.
> > >
> > > **Resp_Table 5.** Precision and recall of the identified bias-conflicting samples by shallow networks and rank-regularized ResNet-18. Second and third columns for the UTKFace (age), and rest for the CelebA (makeup).
> > > | Networks  | Precision (%) | Recall (%) | Precision (%) | Recall (%) |
> > > |-----------|---------------|------------|---------------|------------|
> > > | Shallow   | 55.68         | 69.98      | 27.49         | **33.79**      |
> > > | ResNet-18 | **68.67**         | **75.94**      | **55.29**        | 32.46      |
> > >
> > > ___
> > > ### References.
> > >
> > > [1] Bardes, Adrien, Jean Ponce, and Yann LeCun. "Vicreg: Variance-invariance-covariance regularization for self-supervised learning." ICLR 2022
> > >
> > > [2] Zbontar, Jure, et al. "Barlow twins: Self-supervised learning via redundancy reduction." ICML 2021.
> > >
> > > [3] Rahaman, Nasim, et al. "On the spectral bias of neural networks." ICML 2019
> > >
> > > [4] Neyshabur, Behnam, Ryota Tomioka, and Nathan Srebro. "In search of the real inductive bias: On the role of implicit regularization in deep learning." ICLR 2015

---

> ### Comment · Reviewer_datJ · 2022-11-21
> **Appreciate authors' response and Raise score**
>
> I have read the rebuttal. It fixes most of my concerns. I appreciated the new experiments for Q4 and Q5. Thus, I decided to raise the score from 5 to 6.
>
> Figure 3 (a) is interesting, as it somehow showed that the color (spurious feature) is easy-to-learn. Also, Figure 3 (c) supports the hypothesis of the proposed methods. The paper can be stronger if the authors can show similar phenomena in CelebA or Waterbirds.

---

> > ### Author Response · Authors · 2022-11-22
> > **Thank you!**
> >
> > We sincerely appreciate you taking the time to provide your insightful feedback and raising the score. We are certain your insights and the discussion have improved our work. We will also try our best to incorporate your suggestion into the next version of our paper.

---

### Official Review · Reviewer_rRxe · 2022-10-28

**Confidence:** 4
**Correctness:** 4
**Technical Novelty And Significance:** 4
**Empirical Novelty And Significance:** 3
**Recommendation:** 8

**Clarity, Quality, Novelty And Reproducibility:**

The paper is very clear, and engages the reader in a clear motivational narrative supported by convincing experiments in Sec. 2. The approach appears to be quite novel. There may be sufficient detail to reproduce the results, but there could be some gaps.

**Strength And Weaknesses:**

Strengths

The goal of the work is to produce an unbiased classifier without having any labels of attributes or classes, which are typically required for debiasing methods. In this problem setup, the method must determine the relevant attributes and whether they are biased, jointly and without any labels i.e. fully unsupervised. This is a very challenging and interesting problem setup that builds upon the recent breakthroughs in unsupervised representation learning.

The method is based on the observation that model bias seems to be inversely correlated with the entropy of the representation learned by the model. The hypothesis is that spurious correlations in training data lead to a concentration of model attention on a few features, leading to poor generalization performance.

The approach seems quite novel. Other popular methods are based on GANs or other adversarial techniques, rather than unsupervised learning of bias. The idea of creating a highly biased encoder, based on rank analysis, is clever and seems to be supported by the results.
The experiments underlying Fig. 1 are very useful to illustrate the insight and motivation underlying the approach. Using controlled settings in simplified datasets is helpful here to illustrate the best-case scenarios that provide evidence of the theory. Many papers lack such intermediate results and rely solely on improved results of the end task to validate elements of the approach.

The experiments are performed on face datasets using combinations of standard attributes including gender, age, race, heavy_makeup that have been annotated previously. A comprehensive set of baselines are compared against, including self-supervised methods, supervised debiasing methods and algorithms that select training sets for debiasing. The proposed method handily outperforms other unsupervised methods, and supervised methods in some cases, which is impressive. The experimental protocol follows that of recent works for consistency.

Weaknesses

In fig. 1 the bias ratios seem very high, 95% or greater, and the plot shows that the amount of change from 0 to 95% is only slightly larger than the change from 95 to 98%, i.e. there is little change in the effective rank when the ratio ranges from 0 to close to 95%. At face value this seems to reveal a significant disadvantage in using the effective rank as it is insensitive to the bias ratio until it becomes extreme. Is that the case?

Is the matrix Z in sec. 2.3, formed from the input vectors and their corresponding feature vectors, the matrix that is used in the rank computations in Fig. 1? If so this should be stated earlier, in the discussion of Fig. 1, with a forward reference. As written the paper is unclear on what the rank of a representation might be until sec. 2.3.

There are important related works in discovering biases without attribute labels that are not cited:

Lang, O., Gandelsman, Y., Yarom, M., Wald, Y., Elidan, G., Hassidim, A., Freeman, W.T., Isola, P., Globerson, A., Irani, M., Mosseri, I. “Explaining in Style: Training a GAN to explain a classifier in StyleSpace.” ICCV 2021.

Li, Z., Xu, C. “Discover the Unknown Biased Attribute of an Image Classifier.” ICCV 2021.

Another one, in ECCV 2022, published after the ICLR submission deadline:
“Discover and Mitigate Unknown Biases with Debiasing Alternate Networks”

These papers all rely on GANs or other adversarial techniques, and rely on target labels to some extent. Hence they do not infringe on the novelty of the proposed method. However, the discussion of related work should be expanded to include these, and to better differentiate the novel claims.


**Summary Of The Paper:**

The paper proposes a self-supervised method for debiasing models by building a highly biased model, then leveraging that model to select the training set to create an unbiased model. The method does not require any labels of the bias attributes, as many methods do, or even the target classes or attributes, which makes it quite novel. The experiments show that the method outperforms other unsupervised algorithms, even those leveraging class labels.

**Summary Of The Review:**

The paper presents a novel solution to the difficult problem of self-supervised debiasing, an important problem that has received little attention so far. The writing is clear, and the experiments are thorough and convincing. The paper should generate significant interest for those working in bias mitigation.

---

> ### Author Response · Authors · 2022-11-16
> **Official Comment by Paper3840 Authors**
>
> We appreciate your insightful and constructive comments that help us to improve our paper. Here we have tried our best to fully clarify all the issues in the point-by-point responses (the reviewer’s comment is highlighted, followed by our reply).
>
> > **Q1.** There is little change in the effective rank when the ratio ranges from 0 to close to 95%. At face value this seems to reveal a significant disadvantage in using the effective rank as it is insensitive to the bias ratio until it becomes extreme. Is that the case?
>
> Thank you for the insightful comments. While the effective rank may seem insensitive to the bias ratio itself, bias-conflict or unbiased test accuracy can synergistically explain the trends of rank reduction. The unbiased test accuracy for fig. 1 is provided in Table 7 of the revised paper. For example, by fitting linear regression with effective rank (y) in fig. 1c and unbiased test accuracy (X) of CMNIST in Table 7, one can obtain y=0.0106*X + 3.1736, with p-value < 0.005 for the slope coefficient. Similar linear correlations between effective rank and bias-conflict accuracy can be observed in Figure 7 of the appendix.
>
> Moreover, we have additionally applied the rank regularization to the moderately biased CMNIST, e.g., bias ratio=60%. Resp_Table 1 below shows that rank regularization works well in a natural setting. In addition, Waterbirds in Figure 2c, UTKFace and CelebA correspond to a more general dataset with higher resolution and a moderate bias ratio. These observations suggest that the effective rank, bias ratio, and unbiased accuracy are closely related.
>
>
> **Resp_Table 1.** Rank regularization with moderate level of bias (CMNIST, bias ratio=60%).  $\lambda_{reg}=50$ is used to train the rank-regularized model. Bias-aligned accuracy, bias-conflict accuracy, precision and recall of the identified bias-conflicting samples are reported.
> | Methods | Align (%) | Conflict (%) | Precision (%) | Recall (%) |
> |---------|---------|----------|-----------|--------|
> | ERM     | 99.49   | 97.81    | 79.55     | 0.87   |
> | Ours    | 96.25   | 38.15    | 91.56     | 60.97  |
>
> > **Q2.** Is the matrix Z in sec. 2.3, formed from the input vectors and their corresponding feature vectors, the matrix that is used in the rank computations in Fig. 1? If so this should be stated earlier, in the discussion of Fig. 1, with a forward reference.
>
> To avoid potential confusion, we have added explanations and forward inference in figure 1.
>
> > **Q3.** There are important related works in discovering biases without attribute labels that are not cited.
>
> We found our work to be relevant to the suggested works. We thank the Reviewer for providing important references. We add discussions of these works in the related work section of the revised paper. Discovering and visualizing the unknown biases provide valuable insights into the decision-making of the neural networks. Lang, O, et al. proposes a novel framework, StylEx, which trains a styleGAN to specifically visualize multiple attributes underlying the classifier decisions. Li, Z, et al. obtains a biased attribute hyperplane of the generative models, which can help identify semantic biases by generating bias-traversal images. Li, Z, et al. introduces the discoverer model, which uncovers multiple unknown biases such that the difference of averaged predicted probabilities on the target attribute in two groups is maximized. While substantial advances have been made in this regard, these works require classifier pretrained with target labels. Extending these adversarial frameworks in unsupervised/self-supervised settings would be an exciting direction for future work.

---

### Author Response · Authors · 2022-11-16
**Official Comment by Paper3840 Authors**

We sincerely thank all the reviewers for their constructive and thorough reviews. We are encouraged that the reviewers think that our paper is “very clear and should generate significant interest for those working in bias mitigation” (rRxe), “has a clear motivation, insights and good performance on some complex datasets” (datJ), “well motivated and ablation studies are designed carefully” (cTbQ), and “study on the effective rank is interesting and possibly impactful” (3q4H). Per all the reviewer’s requests, we included several changes in our revised paper:

- Clarified contributions in the introduction, corrected typos and errors, added missing experimental details and error bars.
- More discussions on **related works** including ones suggested by rRxe and 3q4H (Appendix B)
- **Visualization** of the representations with varying degrees of bias (Appendix C.1)
- Comparisons of **minority mining performance** in the supervised setting (Appendix C.1)
- Observed rank reduction trends in **Vision Transformer** (Appendix C.1)
- Ablation study of rank regularization with a **moderate level of bias** (Appendix C.1)
- Additional results on **CelebA (blonde)** (Appendix C.2)
- Examination of the potential of **shallow networks** as the bias-capturing model (Appendix C.4)

More details and explanations on the questions are elaborated in the following responses.

---

### Decision · Program_Chairs · 2023-01-20

**Decision:**

Reject

**Justification For Why Not Higher Score:**

The paper's hypothesis of using SSL as a pre-training strategy is interesting, but comes with a set of conceptual implications. The possibilities and properties of this SSL pre-training were not exploited well enough. Moreover, the comparison in Table 2 seems a bit unfair to "fully supervised" models, as the labels are used anyway when training the final model. In that setup, the inputs / outputs of the whole system should be taken into account when deciding what to put in bold or not.

**Justification For Why Not Lower Score:**

N/A

**Metareview: Summary, Strengths And Weaknesses:**

This paper proposes a procedure for robustifying classification models, by reducing their dependence on spurious correlations. The authors suggest using rank-regularization + SSL pre-training to implement a bias model, then training a "main" model by re-weighting data based on the bias model.
While tackling an interesting problem, the paper lacks in clarity and has some conceptual loopholes. First of all, one of the promises of SSL is that very general models could be trained on large collections of data. Here, the bias model trained with SSL is trained on the task data specifically and general pre-trained SSL models are not explored. Second, the results presented in Table 2 fall short behind JTT and ERM. The proposed method does not require labels when training the bias model (hence the cross), but that does not really justify not bolding JTT, as the main model is trained on labeled data anyway.
The paper received diverging reviews, but most reviewers agreed that the paper could further be improved based on the reviewer's feedback. Given the reviews, rebuttal, discussion and AC/R meeting, I recommend this paper for rejection, and encourage the authors to improve the manuscript and submit to another venue.

**Summary Of Ac-Reviewer Meeting:**

During the meeting, we discussed the paper's strengths and weaknesses. While all reviewers agreed that the tackled problem is interesting, some conceptual flaws were pointed out. Reviewers with a borderline score, agreed that these flaws, together with the lack of clarity in the contributions and experimental setup could grant rejection.
We concluded the meeting by saying that the paper could definitely improved and once improved would constitute a good research artefact.